# Negative correlation of single-cell *PAX3:FOXO1* expression with tumorigenicity in rhabdomyosarcoma

Carla Regina[1], Ebrahem Hamed[1], Geoffroy Andrieux[2,3,4], Sina Angenendt[1], Michaela Schneider[1], Manching Ku[1], Marie Follo[5], Marco Wachtel[6], Eugene Ke[7], Ken Kikuchi[8], Anton G Henssen[9], Beat W Schäfer[6], Melanie Boerries[2,3,4,13], Amy J Wagers[10,11,12], Charles Keller[14], Simone Hettmer[1,13,15]

Rhabdomyosarcomas (RMS) are phenotypically and functionally heterogeneous. Both primary human RMS cultures and low-passage *Myf6Cre,Pax3:Foxo1,p53* mouse RMS cell lines, which express the fusion oncoprotein Pax3:Foxo1 and lack the tumor suppressor *Tp53* (*Myf6Cre,Pax3:Foxo1,p53*), exhibit marked heterogeneity in *PAX3:FOXO1* (*P3F*) expression at the single cell level. In mouse RMS cells, *P3F* expression is directed by the *Pax3* promoter and coupled to *eYFP*. YFP[low]/P3F[low] mouse RMS cells included 87% G0/G1 cells and reorganized their actin cytoskeleton to produce a cellular phenotype characterized by more efficient adhesion and migration. This translated into higher tumor-propagating cell frequencies of YFP[low]/P3F[low] compared with YFP[high]/P3F[high] cells. Both YFP[low]/P3F[low] and YFP[high]/P3F[high] cells gave rise to mixed clones in vitro, consistent with fluctuations in *P3F* expression over time. Exposure to the anti-tropomyosin compound TR100 disrupted the cytoskeleton and reversed enhanced migration and adhesion of YFP[low]/P3F[low] RMS cells. Heterogeneous expression of *PAX3:FOXO1* at the single cell level may provide a critical advantage during tumor progression.

## Introduction

Rhabdomyosarcoma, the most common soft tissue sarcoma in children and adolescents, comprises two main genotypes defined by the presence or absence of *PAX* gene rearrangements (1, 2). Canonical *PAX* translocations juxtapose the N-terminus of the paired-box genes *PAX3* or *PAX7* with the C terminus of the transcription factor *FOXO1* (3). *PAX3:*

*FOXO1* (*P3F*) has been detected in 55% and *PAX7:FOXO1* (*P7F*) in 22% of alveolar histology RMS tumors (4). Both *PAX* gene fusions act as major oncogenic drivers. *P3F* was shown to cooperate with the master transcription factors *MYOG*, *MYOD*, and *MYCN* to recruit superenhancers and establish autoregulatory loops that enforce its myogenic and oncogenic transcriptional program (5). *P3F* knockdown in human and mouse RMS cell lines was linked to a decrease in proliferation rates (6, 7). Patients with RMS harboring *P3F* are more likely to present with metastatic disease and relapse quickly despite aggressive therapy. Extremely poor survival rates call for a deeper understanding of the biology of *P3F*+ RMS (2).

Several independent studies confirm that ectopic *P3F* alone does not induce RMS tumors in mice (14, 8). Additional oncogenetic hits are necessary to initiate P3F-expressing myogenic tumors from cells of both myogenic and non-myogenic lineage (9, 15), provided that ectopic *P3F* is expressed before the introduction of these additional oncogenic events (16). This observation is consistent with genomic subclonality analyses in human tumors identifying *P3F* as a founding event in P3F+ RMS (11). *PAX*-translocated RMS tumors have extremely low overall mutation rates (10), but they tend to harbor regions of genomic amplification, often involving the proto-oncogene *MYCN*, the cell cycle regulator *CDK4*, and the *TP53* pathway modulator *MDM2* (11, 12). Indeed, transcriptional profiling indicated widespread inactivation of *TP53* signaling in P3F+ RMS (13). Keller et al established a mouse model of *P3F*-expressing alveolar RMS by combining conditional activation of biallelic *P3F* expression from the endogenous *Pax3* locus and homozygous deletion of *Tp53* in *Myf6*-expressing maturing mouse myofibers (14). In this system, *P3F* is linked to an *eYFP* fluorescent marker gene, which is expressed as a second cistron downstream from an internal ribosome entry site (IRES) on the same mRNA, so that *P3F* and

[1]Division of Pediatric Hematology and Oncology, Department of Pediatric and Adolescent Medicine, University Medical Center Freiburg, University of Freiburg, Freiburg, Germany   [2]Institute of Medical Bioinformatics and Systems Medicine, Medical Center–University of Freiburg, Faculty of Medicine, University of Freiburg, Freiburg, Germany   [3]German Cancer Consortium (DKTK), Partner Site Freiburg, Germany   [4]German Cancer Research Center (DKFZ), Heidelberg, Germany   [5]Department of Medicine I, Medical Center - University of Freiburg, Faculty of Medicine, University of Freiburg, Freiburg, Germany   [6]University Children's Hospital, Children's Research Center and Department of Oncology, Zürich, Switzerland   [7]Department of Microbiology, Immunology and Cancer Biology, School of Medicine, University of Virginia, Charlottesville, VA, USA   [8]Department of Pediatrics, Graduate School of Medical Science, Kyoto Prefectural University of Medicine, Kyoto, Japan   [9]Experimental and Clinical Research Center of the Max Delbrück Center and Charité Berlin, Berlin, Germany   [10]Department of Stem Cell and Regenerative Biology, Harvard University, Harvard Stem Cell Institute, Cambridge, MA, USA   [11]Joslin Diabetes Center, Boston, MA, USA   [12]Paul F. Glenn Center for the Biology of Aging, Harvard Medical School, Boston, MA, USA   [13]Comprehensive Cancer Centre Freiburg, Medical Center–University of Freiburg, Freiburg, Germany   [14]Children's Cancer Therapy Development Institute, Beaverton, OR, USA   [15]Spemann Graduate School of Biology and Medicine (SGBM), Freiburg, Germany

Correspondence: simone.hettmer@uniklinik-freiburg.de

*YFP* expression strongly correlate and YFP fluorescence can be used as a surrogate for *P3F* transcription from the *Pax3* locus ([17], [15]). YFP activity in such *Myf6Cre+/−,Pax3:Foxo1+/+,p53−/−* mouse RMS tumors was previously shown to differ between individual tumor cells and fluctuate over time, consistent with heterogeneous and dynamic expression of *P3F* at the single cell level ([18]). This study aimed to clarify the functional impact of variable *P3F* expression at the cellular level in *Myf6Cre+/−,Pax3:Foxo1+/+,p53−/−* mouse RMS tumors. Our observations reveal higher tumor-propagating potential of P3F[low] cell states than P3F[high] cell states.

# Results

### Variable cellular *P3F* dose in mouse and human RMS cells

In mouse U23674 and U21459 cells (established from *Myf6Cre+/−, Pax3:Foxo1+/+,p53−/−* mouse sarcomas), expression of *P3F* is directed by the *Pax3* promoter and coupled to an *eYFP* fluorescent marker, which is activated as a second cistron downstream from an encephalomyocarditis virus–derived IRES ([17]). As previously described ([18]), the U23674 and U21459 cell pools are composed of cells expressing different YFP levels. To explore whether similar cell-to-cell variability of *P3F* transcript levels may also occur in human RMS, single-cell digital droplet PCR was performed to quantify the absolute number of *P3F* and *GAPDH* mRNA molecules per single cell in three human RMS patient–derived primary cell cultures (IC-pPDX-35, RMSZH003, and SJRHB013759_X1) and in two human cell lines (Rh41 and Rh30) ([Figs 1A and B] and [S1B] and Table S1). *P3F* was detected in 41 of 83 (49%) *GAPDH*-expressing IC-pPDX-35 cells, 17 of 79 (22%) *GAPDH*-expressing RMSZH003 cells, 21 of 87 (24%) *GAPDH*-expressing SJRHB013759_X1 cells, 53 of 85 (62%) *GAPDH*-expressing Rh41 cells, and 12 of 46 (26%) *GAPDH*-expressing Rh30 cells ([Fig 1A]). Normalization of *P3F* expression based on *GAPDH* expression highlighted that cells with equivalent numbers of *GAPDH* mRNA molecules displayed substantial variation in *P3F* expression ([Fig 1B and C]). We conclude that, similar to what was observed in *Myf6-Cre,Pax3:Foxo1,p53* mouse RMS tumors ([18]), there is substantial cell-to-cell variability in *P3F* expression in the human RMS cell pool.

### Fluctuation of *P3F* expression in mouse RMS cells between P3F[high] and P3F[low] states

To further investigate the behavior of RMS cells expressing different P3F levels, U23674 cells were subfractionated by fluorescence-activated cell sorting (FACS) to discriminate YFP[high] (Y-H) and YFP[low] (Y-L) cell subsets ([Fig S1A]; purity of sorted populations >98%), with gates determined based on fluorescence detection in YFP-negative and YFP-positive control samples. On average, U23674 cells contained 26% ± 8.5% YFP[high] and 74% ± 8.5% YFP[low] cells. RT-qPCR ([Fig 2A]) and Western blot ([Fig 2B]) confirmed that the YFP[high] (Y-H) subset of U23674 cells expressed higher levels of *P3F* than significantly lower *P3F* levels in YFP[low] (Y-L) and unfractionated (U) U23674 cells, and absent *P3F* in normal skeletal muscle (SM, [Fig 2A]). For each U23674 cell subset (i.e., unfractionated, YFP[high]/P3F[high] or YFP[low]/P3F[low] cells), 20 cells per well were plated into 96-well

plates ([Fig 2C]). Clones formed in 291 of 1,220 (23.9%) YFP[low]/P3F[low] U23674 cells, 64 of 1,220 (5.2%) YFP[high]/P3F[high] U23674 cells, and 114 of 1,220 (9.3%) unfractionated U23674 cells ([Fig 2D]; *P* < 0.001). These differences were confirmed in three independent experiments. Thus, surprisingly, YFP[low]/P3F[low] U23674 cells exhibited higher clonal activity in vitro than YFP[high]/P3F[high] U23674 cells.

Clones originating from unfractionated, YFP[high]/P3F[high] and YFP[low]/P3F[low] U23674 cells were allowed expansion for up to 11 d ([Fig 2C]). Average expression of *P3F* mRNA in day 11 (d11) clones originating from unfractionated, YFP[high]/P3F[high] and YFP[low]/P3F[low] U23674 cells was determined by RT-QPCR and found to be similar ([Fig 2E]). The proportion of YFP[high]/P3F[high] and YFP[low]/P3F[low] cells in d11 clones was further analyzed by FACS. Interestingly, all d11 clones, including those arising from YFP[low]/P3F[low] cells, contained both YFP[low]/P3F[low] and YFP[high]/P3F[high] cells. Specifically, there were 13.2% ± 7.4% YFP[high]/P3F[high] cells in clones originating from unfractionated cells compared with 27.5% ± 26.2% YFP[high]/P3F[high] cells in clones originating from YFP[high]/P3F[high] cells (*P* = 0.93) and 7.9% ± 2.6% YFP[high]/P3F[high] cells in clones originating from YFP[low]/P3F[low] cells (*P* = 0.54; [Fig 2F]). Thus, remarkably, when cultured in vitro, YFP[low]/P3F[low] U23674 cells gave rise to clones containing YFP[high]/P3F[high] as well as YFP[low]/P3F[low] cells and vice versa. This observation is consistent with dynamic expression of *P3F* in single U23674 cells, as previously reported by Kikuchi et al ([18]).

For comparison, we evaluated the dynamics of expression in myogenic cells of a fluorescent protein that was not linked to P3F expression. C2C12 mouse myoblasts were transduced with empty vector (EV) pMSCV-Flag-IRES-GFP retroviruses, and EV-GFP[high] and EV-GFP[low] cells were separated by FACS sorting (purity > 98%) ([Fig S2A]). Twenty sorted EV-GFP[high] and EV-GFP[low] cells per well were plated into 96-well plates. Clones were allowed to expand for 10 d and analyzed by FACS. In contrast to results obtained when YFP was coupled to P3F expression ([Fig 2]), EV-GFP–expressing cells showed stable fluorescence phenotypes and similar clonal efficiencies. In particular, clonal efficiency was 6.3% for EV-GFP[high] and 6.4% for EV-GFP[low] cells ([Fig S2B]). Day 10 (d10) clones arising from EV-GFP[high] cells contained 99% ± 1.3% GFP[high] cells, whereas d10 clones originating from EV-GFP[low] cells contained 0% ± 0% GFP[high] cells ([Fig S2C and D]). These data provide further support to the notion that dynamic expression patterns seen in cells expressing YFP coupled to P3F reflect variation in P3F expression.

### Changes in P3F dose in mouse RMS cells in response to environmental changes

To examine if changes in the cell environment influenced the proportion of YFP[high]/P3F[high] and YFP[low]/P3F[low] U23674 cells, we measured the percentage of YFP[high]/P3F[high] cells in U23674 and U21459 cells grown in medium containing different glutamine and glucose concentrations ([Figs S3A and B] and [S4A and B]), plated at different densities ([Figs S3C, D, and F–H] and [S4C]), cultured on surfaces covered with different matrices ([Figs S3E] and [S4F]) and exposed to different drugs ([Figs S3I–K] and [S4D and E]). We observed that higher glutamine levels led to an increase in the percentage of YFP[high]/P3F[high] U23674 cells (2% ± 0.3% versus 15% ± 0.8% YFP[high]/P3F[high] cells in medium containing 0.05 mM versus 4 mM glutamine, *P* = 0.001, [Fig S3A]) and U21459 cells (2.3% ± 0.1% versus 4.4% ± 0.2%

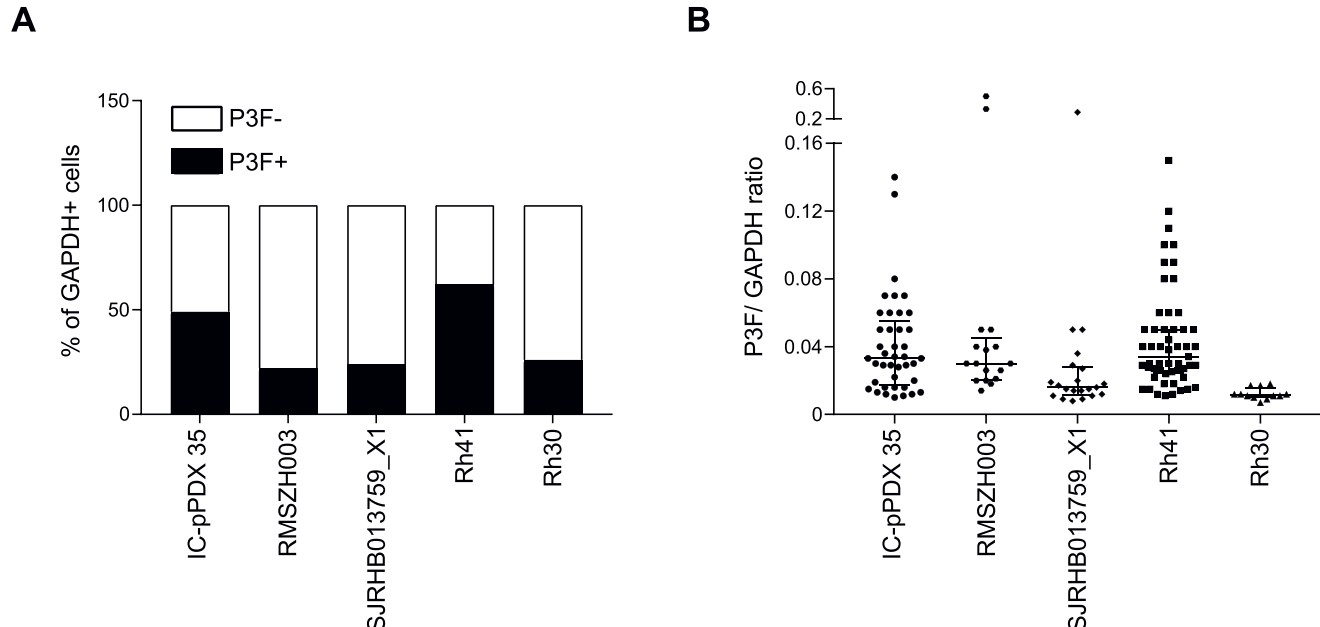

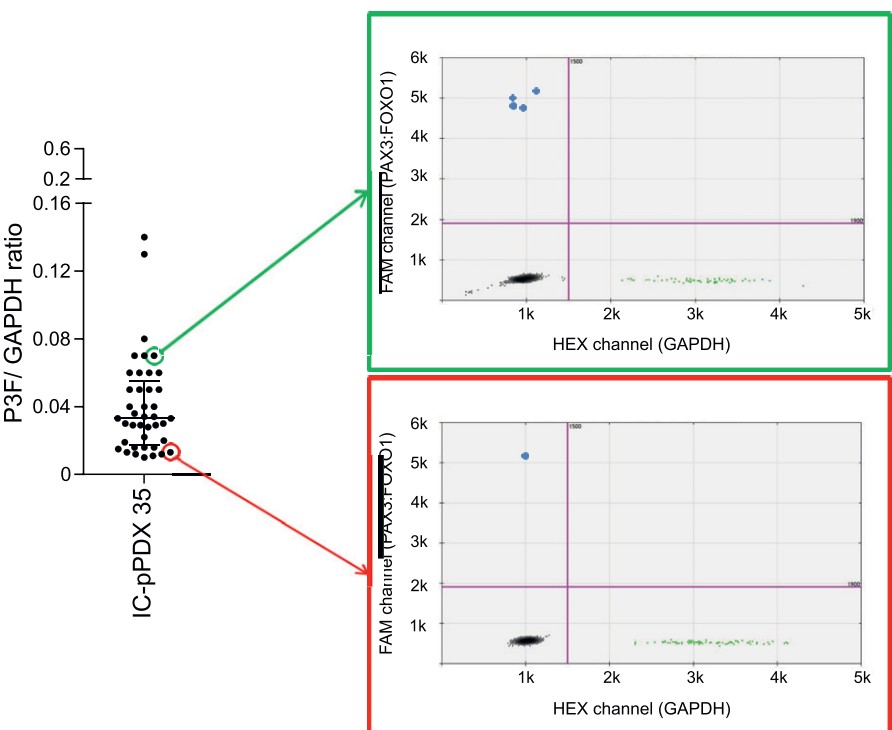

**Figure 1. Cell-to-cell variability in *P3F* expression in RMS.**
**(A, B)** Evaluation of *P3F* and *GAPDH* mRNA expression at the single-cell level by RT-digital droplet PCR in IC-pPDX-35 (n = 88 cells), RMSZH003 (n = 88 cells), and SJRHB013759_X1 (n = 88 cells) human patient-derived RMS cell cultures as well as Rh41 (n = 88 cells) and Rh30 (n = 48 cells) human RMS cell lines. **(A)** *P3F* expression was detected in 22–49% of *GAPDH*-expressing cells in patient-derived RMS cell cultures and in 26–62% of *GAPDH*-expressing cells in human RMS cell lines. **(B)** Remarkable cell-to-cell heterogeneity in *P3F* expression as evidenced by representation of *P3F*/*GAPDH* ratios. Each dot represents the mRNA content of one cell. Bars indicate medians ± interquartile ranges. **(C)** 2D representation of droplets generated from two IC-pPDX-35 cells with P3F$^{high}$ (upper panel, marked in green) or P3F$^{low}$ (bottom panel, marked in red) profiles with the same level of *GAPDH* mRNA. Please see Fig S1B for single-cell *P3F* and *GAPDH* expression in each line.

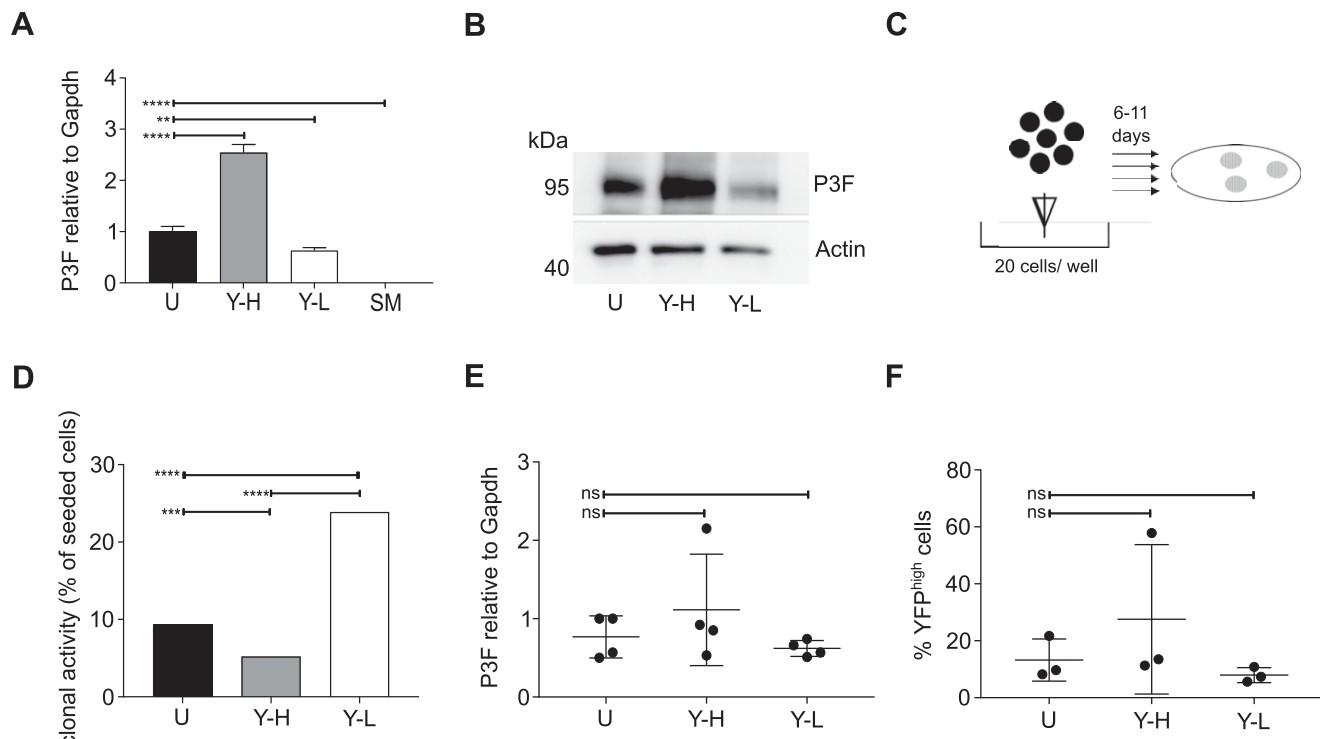

**Figure 2. Higher clonogenic activity of YFP$^{low}$/P3F$^{low}$ than YFP$^{high}$/P3F$^{high}$ mouse U23674 RMS cells.**
The mouse RMS cell line U23674 was sorted into YFP$^{high}$/P3F$^{high}$ and YFP$^{low}$/P3F$^{low}$ cells (purity > 98%). **(A, B)** RT-QPCR and (B) WB demonstrated enrichment of P3F in the YFP$^{high}$ (Y-H) compared with the YFP$^{low}$ (Y-L) and unfractionated (U) cell subsets. Skeletal muscle cells served as P3F$^{neg}$ control cells. **(C, D)** Clonally sorted YFP$^{low}$/P3F$^{low}$ cells exhibited significantly higher clonal activity (291 [23.9%] clones out of 1,220 cells plated) than YFP$^{high}$/P3F$^{high}$ (64 [5.2%] clones out of 1,220 cells plated) and unfractionated U23674 cells (114 [9.3%] clones out of 1,220 cells plated). **(C, E, F)** The composition of clones arising from unfractionated, YFP$^{high}$/P3F$^{high}$ and YFP$^{low}$/P3F$^{low}$ U23674 cells was analyzed 11 d after plating by (E) RT-QPCR (four clones per cell subset analyzed) and by (F) FACS (three clones per cell subset analyzed): (E) *P3F* expression levels in clones arising from unfractionated, YFP$^{high}$/P3F$^{high}$, and YFP$^{low}$/P3F$^{low}$ cells were similar. **(F)** Clones arising from unfractionated, YFP$^{high}$/P3F$^{high}$, and YFP$^{low}$/P3F$^{low}$ contained a mix of YFP$^{high}$ and YFP$^{low}$ cells. Differences in clonal activity were evaluated for statistical significance by chi square test; differences in *P3F* expression and cell composition by one-way ANOVAs (****$P$ < 0.001; ***$P$ < 0.01; **$P$ < 0.01; ns $P$ ≥ 0.05). All experiments were replicated three times. Please see Fig S2 for clonal expansion of C2C12 cells transduced with retroviruses expressing GFP. Please see Fig S3 for differences in the proportion of YFP$^{high}$ U23674 cells cultured under different conditions.

YFP$^{high}$/P3F$^{high}$ cells in medium containing 0.05 mM versus 4 mM glutamine, $P$ < 0.0001, Fig S4A). Changes in glucose concentrations in the medium did not affect the percentage of YFP$^{high}$/P3F$^{high}$ U23674 and U21459 cells (Fig S3B).

Differences in the extracellular matrix used for cell culture also influenced the percentage of YFP$^{high}$/P3F$^{high}$ cells. When cells were grown on laminin or Matrigel, the percentage of YFP$^{high}$/P3F$^{high}$ U23674 cells was reduced compared with cells grown on uncoated surfaces (21% ± 0.4% YFP$^{high}$/P3F$^{high}$ cells on laminin and 13% ± 0.6% YFP$^{high}$/P3F$^{high}$ cells on Matrigel versus 32% ± 1.2% YFP$^{high}$/P3F$^{high}$ cells on uncoated surfaces; $P$ < 0.001, Fig S3E). Exposure to fibronectin did not alter the percentage of YFP$^{high}$/P3F$^{high}$ U23674 cells (Fig S3E). Exposure to Matrigel, but not laminin or fibronectin, also reduced the percentage of YFP$^{high}$/P3F$^{high}$ U21459 cells compared with culture uncoated surfaces (2.1% ± 0.4% YFP$^{high}$/P3F$^{high}$ cells on Matrigel versus 3.9% ± 0.6% YFP$^{high}$/P3F$^{high}$ cells on uncoated surfaces; $P$ < 0.05, Fig S4F). Moreover, culture at higher cell densities, achieved by seeding cells in triangle-shaped wells (Fig S3C) or at higher cell numbers per well (Fig S3D and F–H), increased the percentage of YFP$^{high}$/P3F$^{high}$ U23674 and U21459 cells. For example, cells seeded at 30,000 cells per well on uncoated surfaces contained more YFP$^{high}$/P3F$^{high}$ cells than those seeded at 5,000 cells

per well on uncoated surfaces (44% ± 0% versus 25% ± 3.4%; $P$ = 0.001; Fig S3D).

Finally, U23674 and U21459 cells were treated with two chemotherapy drugs used for treatment of RMS (vincristine, dactinomycin; 48 h exposure each) and with the anti-tropomyosin compound TR100 for 12 h. Vincristine raised the percentage of YFP$^{high}$/P3F$^{high}$ U23674 cells (60% ± 0.8% versus 41% ± 1.1% YFP$^{high}$/P3F$^{high}$ cells among vincristine-treated compared with control cells; $P$ = 0.0001; Fig S3I) and U21459 cells (4.5% ± 0.4% versus 3.6% ± 0.2% YFP$^{high}$/P3F$^{high}$ cells among vincristine-treated compared with control cells; $P$ < 0.05; Fig S4D), and dactinomycin decreased the percentage of YFP$^{high}$/P3F$^{high}$ U23674 cells (33.4% ± 3% versus 43.4% ± 1.1%; YFP$^{high}$/P3F$^{high}$ cells among dactinomycine-treated compared with control cells; $P$ = 0.006; Fig S3J) and U21459 cells (3.7% ± 0.4% versus 5.6% ± 0.6% YFP$^{high}$/P3F$^{high}$ cells among dactinomycine-treated compared with control cells; $P$ < 0.05; Fig S4E). Exposure to TR100 did not change the proportion of YFP$^{high}$/P3F$^{high}$ and YFP$^{low}$/P3F$^{low}$ U23674 cells (Fig S3K).

Extended analyses revealed reduced absolute numbers of YFP$^{low}$ and YFP$^{high}$ cells exposed to vincristine and low-glutamine conditions (50,008 ± 0 Vincristine-exposed and 48,791 ± 1,865 glutamine-deprived versus 121,727 ± 1,821 control YFP$^{low}$ cells, $P$ < 0.01 and 25,992 ± 0 vincristine-exposed and 25,210 ± 964 glutamine-deprived versus

68,274 ± 1,016 control YFP[high] cells, $P < 0.01$); dactinomycin exposure only resulted in a trend towards lower absolute numbers of YFP[high] cells (43,146 ± 754 versus 68,274 ± 1,016 control YFP[high] cells, $P$ = ns, Fig S5A and B). The overall distribution of YFP[high] and YFP[low] cells across cell cycle phases remained the same for all conditions (Fig S5A, B, E,

and F). Generally, YFP[low]/P3F[low] cells included more cells in G0/G1 stages compared with YFP[high]/P3F[high] cells (Figs 3D and E and S5A and B). However, changes in absolute cell numbers correlated with higher percentages of apoptotic YFP[low] and YFP[high] cells exposed to vincristine (9.0% ± 1.7% vincristine-exposed versus 4.3% ± 2.8% control

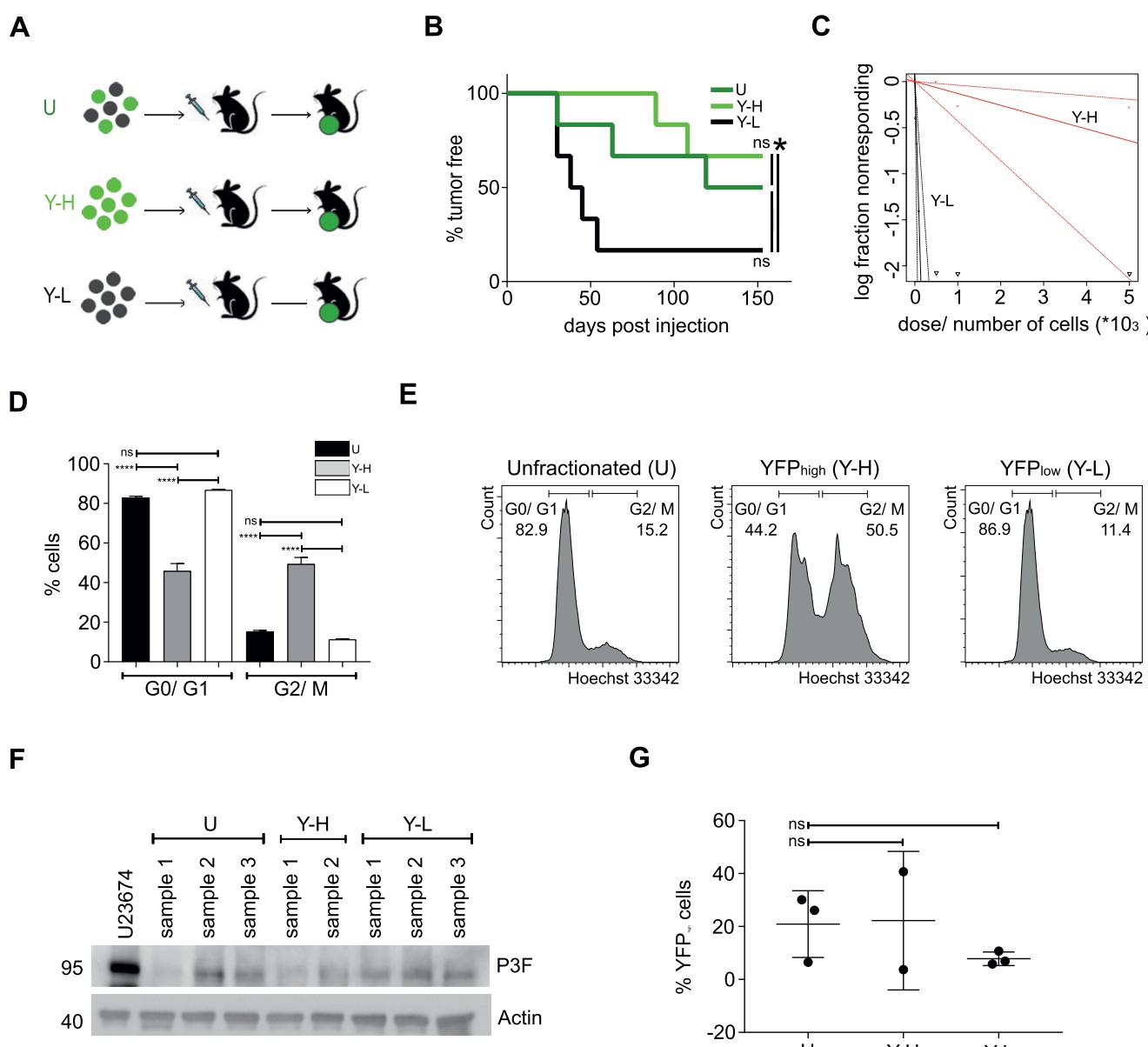

**Figure 3. Higher tumor-propagating activity of YFP[low]/P3F[low] than YFP[high]/P3F[high] mouse U23674 RMS cells.**
**(A)** Unfractionated (U), YFP[high]/P3F[high] (Y-H), and YFP[low]/P3F[low] (Y-L) U23674 cells were implanted into the extremity muscles of NOD.SCID mice and allowed to expand for up to 4 mo. **(B)** YFP[low]/P3F[low] cells formed significantly more tumors at the injection sites than YFP[high]/P3F[high] and unfractionated U23674 cells. **(C)** Limiting dilution analysis revealed significantly higher tumor-repopulating cell frequencies within the YFP[low]/P3F[low] than the YFP[high]/P3F[high] subset of cells: 1 in 7,781 YFP[high]/P3F[high] U23674 cells (95% confidence interval: 1 in 2,332–25,964 cells) versus 1 in 65 YFP[low]/P3F[low] U23674 cells (95% confidence interval: range 1 in 28–151 cells; $P < 0.0001$). **(D, E)** Hoechst 33342 staining determined YFP[low]/P3F[low] cells contained 87% ± 0.2% cells in the G0/G1 phases, and 11% ± 0.3% cells in the G2/M phases of the cell cycle. YFP[high]/P3F[high] cells contained 46% ± 3.7% cells in G0/G1, and 49% ± 3.2% cells in G2/M. **(F, G)** The composition of tumors arising from unfractionated, YFP[high]/P3F[high] and YFP[low]/P3F[low] U23674 cells was analyzed by (F) Western blotting (two to three clones per cell subset analyzed) and by (G) FACS (two to three clones per cell subset analyzed): (F) P3F expression levels in clones arising from unfractionated, YFP[high]/P3F[high], and YFP[low]/P3F[low] cells were similar. **(G)** Tumors arising from unfractionated, YFP[high]/P3F[high], and, notably, YFP[low]/P3F[low] cells contained a mix of YFP[high] and YFP[low] cells. Differences in tumor propagating capacity were evaluated for statistical significance by log-rank (Mantel-Cox) tests; differences in cell composition by ordinary one-way ANOVAs (ns $P \geq 0.05$). Limiting dilution analyses were performed as described by Bonnefoix et al (51) using the limdil function of the StatMod package (author GK Smyth, http://bioinf.wehi.edu.au/software/limdil/). Experiments were replicated three times.

YFP$^{low}$ cells, P < 0.5 and 5.8% ± 0.7% vincristine-exposed versus 2.5% ± 1.5% control YFP$^{high}$ cells, P < 0.001, Fig S5C and D) and a lower percentage of G2/M YFP$^{high}$ cells cultured in low-glutamine conditions (17.0% ± 12.8% glutamine-deprived versus 40.6 ± 3.1 control G2/M YFP$^{high}$ cells, P > 0.01, Fig S5F). Taken together, our observations do not indicate that high P3F levels protect cells from stress-induced apoptosis by inducing a G2/M block.

## Higher tumor-propagating capacity of P3F$^{low}$/YFP$^{low}$ compared with P3F$^{high}$/YFP$^{high}$ mouse RMS cells

As YFP$^{low}$/P3F$^{low}$ U23674 cells formed significantly more clones than YFP$^{high}$/P3F$^{high}$ U23674 cells, we next examined the ability of YFP$^{low}$/ P3F$^{low}$ and YFP$^{high}$/P3F$^{high}$ U23674 cells to form tumors in immuno-compromised mice immediately after sorting. For each U23674 cell subset (i.e., unfractionated, YFP$^{high}$/P3F$^{high}$, and YFP$^{low}$/P3F$^{low}$ cells), 500 cells each were injected into the cardiotoxin preinjured gastrocnemius muscles of NOD.SCID recipients (Fig 3A). Secondary tumors developed in five out of six mice injected with YFP$^{low}$/P3F$^{low}$ U23674 cells, two out of six mice injected with YFP$^{high}$/P3F$^{high}$ U23674 cells, and three out of six mice injected with unfractionated U23674 cells (P < 0.05, Fig 3B). These differences were confirmed in three independent transplantation experiments. Consistent with these observations, limiting dilution analyses (1–5,000 cells implanted in a total of four injections each) revealed that the frequency of tumor-repopulating cells was 1 in 7,781 YFP$^{high}$/P3F$^{high}$ U23674 cells (95% confidence interval: 1 in 2,332 to 25,964 cells) versus 1 in 65 YFP$^{low}$/P3F$^{low}$ U23674 cells (95% confidence interval: range 1 in 28 to 151 cells; P < 0.0001) (Fig 3C and Table S2). Thus, YFP$^{low}$/P3F$^{low}$ U23674 cells clearly exhibited higher tumor-propagating capacity than YFP$^{high}$/P3F$^{high}$ U23674 cells.

It was previously shown that YFP fluorescence in U23674 cells is markedly increased during cell division (18). We therefore used Hoechst 33342 staining to confirm that YFP$^{high}$/P3F$^{high}$ cells contained significantly more cells in the G2/M stages of the cell cycle compared with YFP$^{low}$/P3F$^{low}$ cells (49% ± 3.2% versus 11% ± 0.3%; P < 0.0001, Fig 3D and E). In contrast, YFP$^{low}$/P3F$^{low}$ cells, which exhibited significantly higher tumor-propagating capacity, included more cells in G0/G1 stages compared with YFP$^{high}$/P3F$^{high}$ cells (87% ± 0.2% versus 46% ± 3.7%; P < 0.0001, Fig 3D and E).

P3F/YFP expression in allograft tumor cells was evaluated by Western blot (Fig 3F) and by FACS (Fig 3G). Again, YFP$^{high}$/P3F$^{high}$ cells were detected in allograft tumors originating from unfractionated U23674 cells (20.9% ± 12.6%), YFP$^{high}$/P3F$^{high}$ (22.2 ± 26.2), and YFP$^{low}$/ P3F$^{low}$ U23674 cells (7.8% ± 2.6%) (Fig 3G). Differences in the percentage of YFP$^{high}$/P3F$^{high}$ cells did not reach statistical significance. Infiltrating and surrounding host cells may account for YFP$^{low}$/P3F$^{low}$ cells in allograft tissue arising from transplanted YFP$^{high}$/P3F$^{high}$ cells. However, the presence of YFP$^{high}$/P3F$^{high}$ cells in tumors originating from YFP$^{low}$/P3F$^{low}$ U23674 cells (Fig 3G) further supports dynamic expression of the fusion oncogene in U23674 cells.

## No effect of P3F expression levels on the proportion of apoptotic cells in RMS

Higher P3F levels in tumor cells may be toxic and induce cell death, thereby accounting for lower clonogenic and tumor-propagating capacity of YFP$^{high}$/P3F$^{high}$ U23674 cells. Annexin V (Ann V) staining was used to demonstrate similar rates of living (80% ± 11.5% versus 74.7% ± 8.8%, P > 0.99) and apoptotic (13.4% ± 8% versus 7.6% ± 3.5%, P > 0.99) YFP$^{high}$/P3F$^{high}$ and YFP$^{low}$/P3F$^{low}$ U23674 cells. There was a trend towards lower percentages of necrotic/late apoptotic YFP$^{high}$/P3F$^{high}$ versus YFP$^{low}$/P3F$^{low}$ U23674 cells (5.7% ± 6.1% versus 15.3% ± 6.3%, P = 0.09), but these differences did not reach statistical significance (Fig 4A and B). These observations indicate that differences in the efficiency of tumor and clone formation by YFP$^{high}$/ P3F$^{high}$ U23674 cells are not due to higher rates of apoptosis among cells expressing higher levels of the fusion oncogene. In U23674 cells, YFP and P3F are expressed from the targeted Pax3:Foxo1-ires-eYFP allele (18), which allows for siGFP-induced knockdown of P3F expression (Figs 4C and S6A). As published previously (6, 7), P3F silencing reduced the proliferation rate of U23674 cells (Fig 4D). Yet, similar to data obtained with YFP$^{high}$/P3F$^{high}$ and YFP$^{low}$/P3F$^{low}$ U23674 cells, there were no significant differences in the rates of living (73.7% ± 9% versus 67% ± 13.5%, P = 0.36), apoptotic (8.2% ± 4.4% versus 11.8% ± 5.1%, P > 0.99), and necrotic cells (14.3% ± 5.2% versus 17.1% ± 9.3%, P > 0.99) between U23674 cells transfected with scrambled (Scr) control siRNA or with siGFP (Fig 4E and F). Expression of cleaved caspase 3 (Cl-Casp 3) and cleaved PARP (Cl-Parp) also did not change after Pax3:Foxo1 silencing (Fig 4G).

P3F was also silenced in the high-passage human RMS cell lines Rh30 (Fig S6B) and Rh5 (Fig S6E). Similar to effects in U23674 cells, P3F silencing reduced the proliferation rate of Rh30 cells (Fig S6C) and Rh5 cells (Fig S6F). Expression of cleaved caspase three and cleaved PARP likewise did not change after Pax3:Foxo1 silencing in Rh30 cells (Fig S6D) or in Rh5 cells (Fig S6G).

## Differential regulation of genes involved in ECM-receptor interaction and focal adhesion in YFP$^{low}$/P3F$^{low}$ versus YFP$^{high}$/P3F$^{high}$ mouse RMS cells

To further delineate the underpinnings of differences in clonal activity and tumor-propagating capacity of U23674 cell subsets, the gene expression profiles of YFP$^{high}$/P3F$^{high}$, YFP$^{low}$/P3F$^{low}$, and unfractionated U23674 cells were examined immediately after sorting (Clariom S Assay, mouse; Affymetrix). This analysis revealed profound differences in the transcriptome of YFP$^{high}$/P3F$^{high}$ versus YFP$^{low}$/P3F$^{low}$ and unfractionated U23674 cells (Fig 5A). We focused our subsequent analyses on transcripts that were differentially regulated between YFP$^{low}$/P3F$^{low}$ and YFP$^{high}$/P3F$^{high}$ U23674 cells (Table S3, logFC < −1 or >1, false discovery rate [FDR] < 0.01). The most significantly enriched pathways among these differentially regulated genes included cell junction, plasma membrane region, extracellular matrix, and cell surface (FDR < 0.05, Fig 5B and Table S4). Transcripts involved in focal adhesion (Fig S7) and regulation of the actin cytoskeleton (Fig S8) were differentially expressed in YFP$^{high}$/ P3F$^{high}$ cells versus YFP$^{low}$/P3F$^{low}$ cells. Differentially regulated candidate genes included Integrin α 8 (Itga8), Cadherin 4 (Cdh4), Rho family GTPase 2 (Rnd2), Integrin α 5 (Itga5), Thrombospondin 3 (Thbs3), EGF containing Fibulin Extracellular Matrix Protein 1 (Efemp1), and Laminin Subunit Alpha 5 (Lama5). QRT-PCR in siGFP compared with control and scramble U23674 cells confirmed that lower P3F dose (Fig 5C) was associated with lower levels of Itga8 (Fig 5D) (16, 17) and Cdh4 (Fig 5E) (18), both involved in cell–cell adhesion.

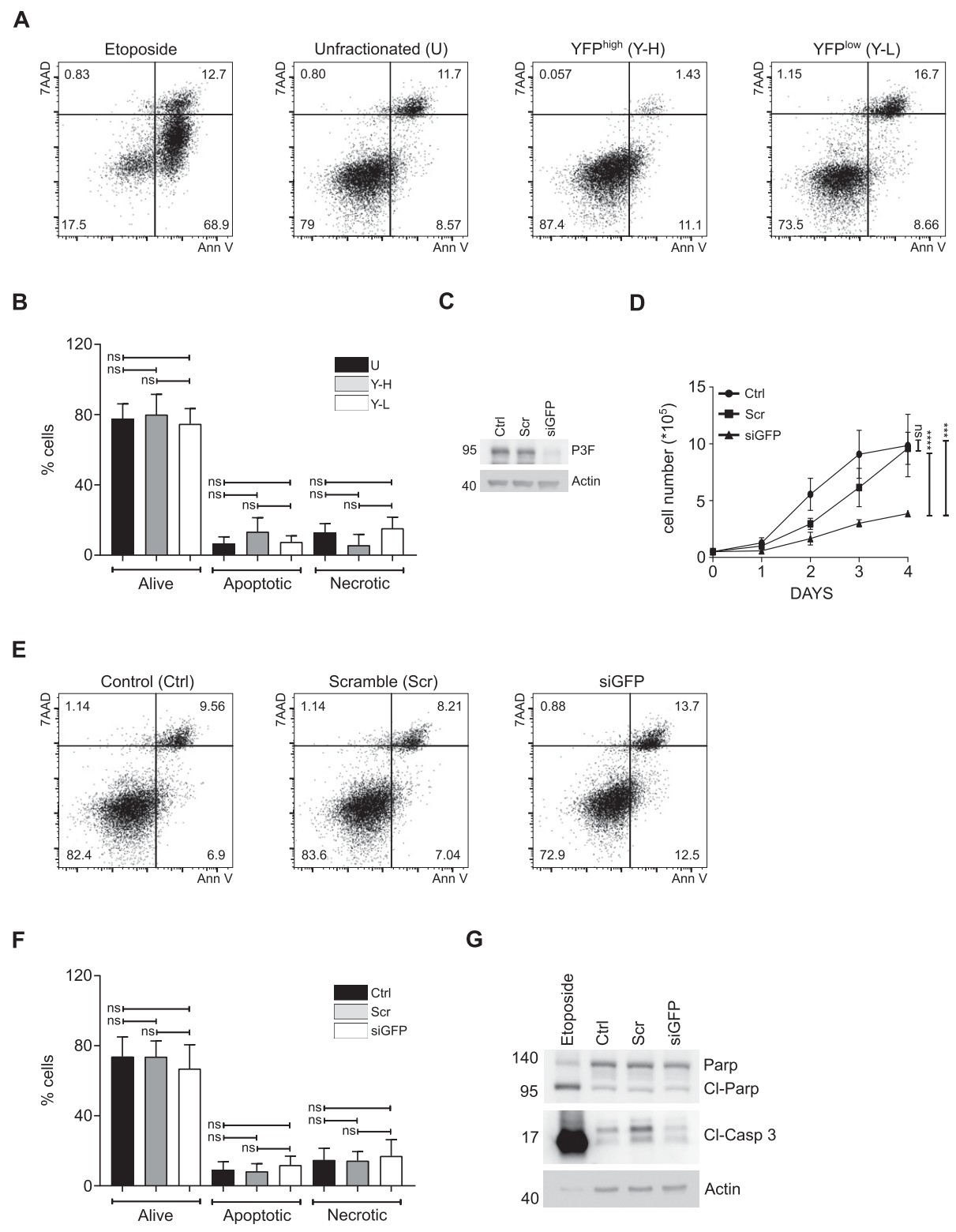

**Figure 4.  Lower P3F dose did not change the proportion of apoptotic cells in mouse U23674 RMS cells stained with Annexin V (Ann V).**
**(A, B)** The proportions of alive Ann V-/7AAD- and apoptotic Ann V+/7AAD-cells among YFP^high/P3F^high (Y-H) and YFP^low/P3F^low (Y-L) U23674 cell subsets were similar.
**(C)** There was a trend towards lower percentages of necrotic/late apoptotic Ann V+/7AAD+ Y-H cells (C) *P3F* silencing in siGFP U23674 mouse RMS cells compared with control (ctrl) cells and cells exposed to scramble (scr) siRNA. **(D)** Reduced P3F expression in si-GFP U23674 cells correlated with lower proliferation rates compared with control and scramble cells. **(E, F)** The proportions of living, apoptotic, and necrotic cells among scramble and siGFP U23674 cells were also similar. **(G)** Levels of cleaved Caspase 3 (Cl-Casp 3) and cleaved PARP (Cl-PARP) were similar among control, scramble, and si-GFP U23674 cells. Etoposide-treated U23674 cells were included as

Conversely, higher levels of *P3F* were associated with higher expression of *Rnd2* (Fig 5F) (19), which regulates organization of the actin cytoskeleton, and with higher expression of *Itga5* (Fig 5G) (20, 21), *Thbs3* (Fig 5H) (22, 23), *Efemp1* (Fig 5I) (24), and *Lama5* (Fig 5J) (25), all involved in adhesion/cell–ECM interaction. Also, differences in expression of myogenic regulatory factors between YFP[high]/P3F[high] and YFP[low]/P3F[low] cells were explored. Significantly higher levels of *myoblast determination protein 1* (*MyoD1*) and a trend towards higher expression of *paired-box transcription factor 7* (*PAX7*) and *myogenic factor 5* (*Myf5*) in YFP[high]/P3F[high] cells were noted (Fig S9).

Next, chromatin accessibility sites in YFP[high]/P3F[high] and YFP[low]/P3F[low] cells were investigated by Assay for transposase-accessible chromatin using sequencing (ATAC-Seq) (26). The *Pax3* promoter region showed higher ATAC-Seq signals in YFP[high]/P3F[high] cells. As *P3F* expression in mouse U23674 cells is directed by the endogenous *Pax3* promoter, this observation is consistent with higher transcription of *P3F* in YFP[high]/P3F[high] cells (Fig 5K). *P3F* fusion protein DNA-binding motif also represented the most significantly enriched motif in YFP[high]/P3F[high] cells (Fig 5L). Overall, YFP[high]/P3F[high] cells contained ~30 times more chromatin accessible sites than YFP[low]/P3F[low] cells (4,592 peaks in YFP[high]/P3F[high] cells versus 142 peaks in YFP[low]/P3F[low] cells, Tables S5 and S6). Gene Set Enrichment Analysis demonstrated that genes involved in gene ontology (GO) categories for cell motility, locomotion, and adhesion again were significantly enriched among those that had stronger ATAC-Seq signals in YFP[low]/P3F[low] cells (Table S7).

### Lower P3F expression changes the cytoarchitecture, adhesion, and migration capacities of mouse RMS cells

Enrichment of genes involved in adhesion among those differentially regulated between YFP[low]/P3F[low] and YFP[high]/P3F[high] cells pointed towards differences in cell adherence and migration capacity. As YFP/P3F expression in individual U23674 cells was unstable and fluctuated over time, we chose to investigate the adhesion capacity of YFP[low]/P3F[low] and YFP[high]/P3F[high] cells using siRNA-mediated down-regulation of P3F expression. Indeed, siGFP U23674 cells, in which P3F levels are suppressed, exhibited higher cell surface areas than control, scramble U23674 cells (1.2 ± 0.8 mm$^2$ versus 0.4 ± 0.2 mm$^2$, *P* < 0.0001, Fig 6A). Also, when equal numbers of siGFP and scramble U23674 cells were allowed to adhere to the surface of a tissue culture plate for 2 h before toluidine blue staining of adherent cells, siGFP U23674 cells displayed significantly higher adhesion capacity than scramble U23674 cells (1.9 ± 0.09 versus 0.6 ± 0.1, *P* < 0.0001; Fig 6B).

Cell spreading and adhesion require the establishment of circumferential adhesion zones along the cell surface and may coincide with changes in cytoarchitecture. U23674 cells were stained using paxillin and phalloidin antibodies to visualize focal adhesion points (Fig 6C and D–F) and actin filaments (Fig 6D, G, and H), respectively. Paxillin staining demonstrated a significantly higher number of paxillin-rich focal adhesion points per cell in siGFP (Fig

6F, lower panel) compared with scramble (Fig 6F, middle panel) U23674 cells (47.5 ± 25.3 versus 8.3 ± 5.8 adhesion points per cell, *P* < 0.01, Fig 6C). Phalloidin staining visualized thin, short actin fibers in scramble U23674 cells (Fig 6H, middle panel), whereas robust actin stress fibers were aligned throughout siGFP U23674 cells (Fig 6H, lower panel). The observation that siGFP U23674 cells exhibited highly organized actin stress fibers (Fig 6H, lower panel) and more focal adhesion points (Fig 6F, lower panel) is consistent with the more effective spreading and adhesion of cells expressing lower P3F doses. As adhesion and spreading capacities of cells impact on their ability to migrate, we also plated U23674 cells in serum-free media on Boyden transwell migration filters to investigate transwell migration. Significantly higher numbers of siGFP (Fig 6I, far right panels) compared with scramble (Fig 6I, middle panels) U23674 cells migrated through pores (98 ± 31.6 versus 43 ± 11.8, *P* < 0.0001, Fig 6I and J).

A second low-passage cell line established from *Myf6Cre,Pax3:Foxo1,p53* mouse sarcoma cells (U21459) was used to confirm that lower P3F dose by siGFP silencing (Fig S10A) correlated with higher cell surface areas (0.75 ± 0.61 mm$^2$ versus 0.2 ± 0.09 mm$^2$, *P* < 0.0001, Fig S10B), more efficient adhesion to plastic surfaces (*P* < 0.0001; Fig S10C), increased numbers of paxillin-rich focal adhesions per cell (24 ± 6.7 versus 4.6 ± 3.6 focal adhesions per cell, *P* < 0.0001, Fig S10D and E–G), increased stretching of the actin cytoskeleton (Fig S10H–J), and higher migration activity (133 ± 15 versus 9 ± 5 migrated cells per well, *P* < 0.0001; Fig S10K and L) compared with scramble cells. Lower *P3F* dose in U21459 cells (Fig S11A) also correlated with lower *Itga8* (Fig S11B), *Cdh4* (Fig S11C), and *Rnd2* (Fig S11D) as well as higher *Itga5* (Fig S11E) and *Thbs3* (Fig S11F) expression.

However, lower P3F dose did not correlate with more efficient adhesion to uncoated plastic surfaces and surfaces covered with collagen I, collagen II, collagen IV, fibronectin, laminin, tenascin, or vitronectin in the high-passage human RMS cell lines Rh5 (Fig S12A and B), Rh30 (Fig S12C and D), and Rh41 (Fig S12E and F). As long-term ex-vivo passage may have introduced artefacts, we also attempted to silence *P3F* expression in human, low-passage primary RMS cell cultures and evaluate differences in adhesion capacity. Unfortunately, these experiments were hampered by substantial toxicity and poor cell survival. After siP3F silencing of fusion oncogene expression in CF1 cells (Fig S13A), we did not detect any differences in adhesion to uncoated plastic surfaces (Fig S13B).

### Reversal of adhesive phenotype of P3F[low] cells by chemical disruption of the actin cytoskeleton in mouse RMS cells

TR100 belongs to a class of anti-tropomyosin compounds, which targets cytoskeletal tropomyosin-containing filaments in cancer cells with high specificity (36) (Fig 7A–O). Increased stretching of the actin cytoskeleton in siGFP U23674 cells (Fig 7O, upper panel) was disrupted by TR100 treatment in siGFP U23674 cells (Fig 7O, lower panel). Also, the increase in paxillin-rich adhesion points observed

---

positive controls. Differences in the percentage of living/apoptotic and necrotic cells were evaluated for statistical significance by ordinary one-way ANOVAs (ns *P* ≥ 0.05). Differences in cell growth were evaluated by nonlinear regression (ns *P* ≥ 0.05, ****P* < 0.001). Experiments were replicated three to five times. See Fig S6 for transient silencing of *P3F* in human Rh30 and Rh5 RMS cells and its effects on proliferation and apoptosis.

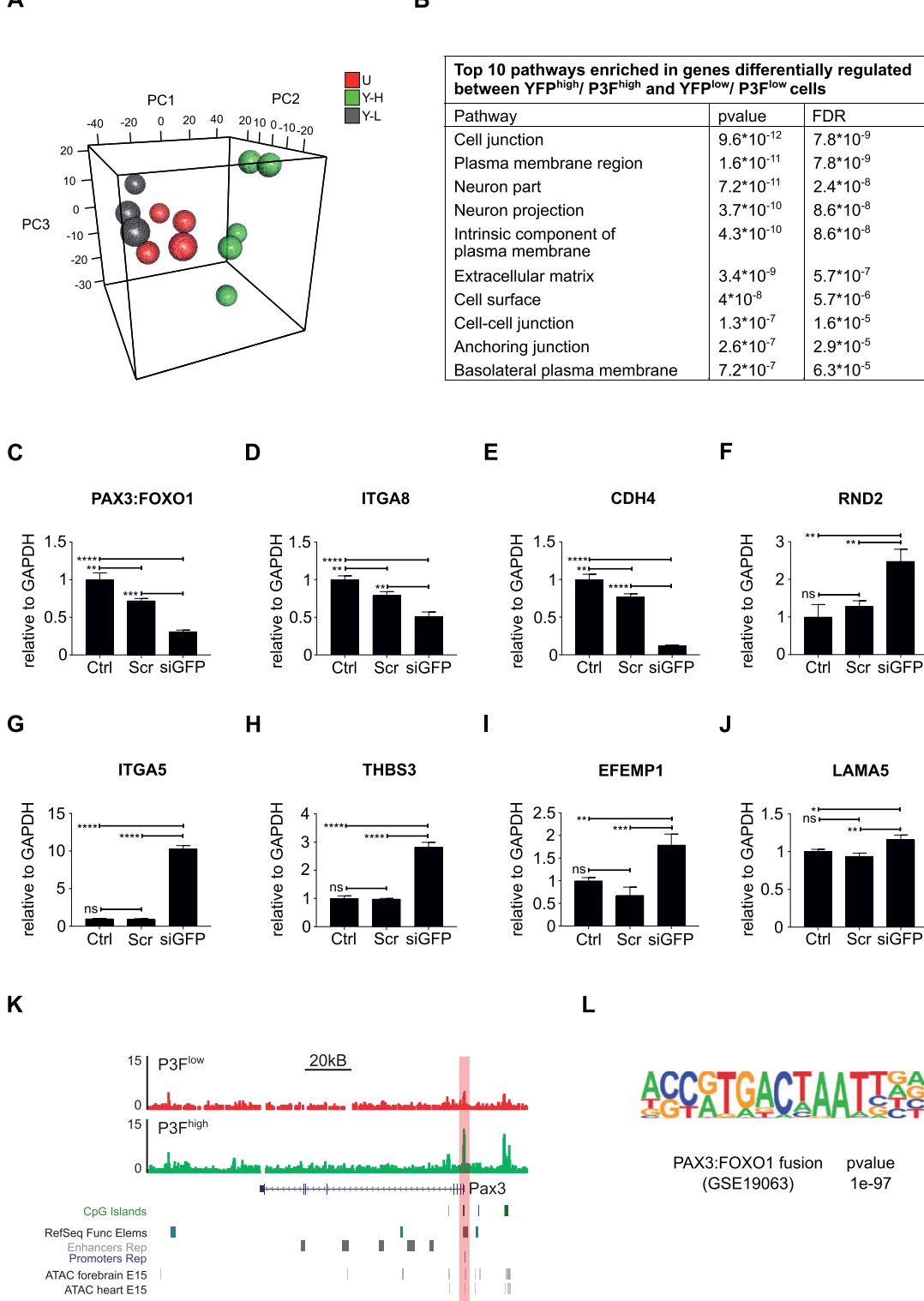

**Figure 5. Transcriptional profiling identifies differences in the expression of genes involved in cell surface/matrix interaction in YFP^high^/P3F^high^ versus YFP^low^/P3F^low^ mouse U23674 RMS cells.**
**(A)** The gene expression profiles of YFP^high^/P3F^high^ and YFP^low^/P3F^low^ cells were distinct as evidenced by principal component analysis. **(B)** Pathway analysis of genes differentially regulated in YFP^high^/P3F^high^ versus YFP^low^/P3F^low^ cells revealed enrichment of transcripts involved in cell junction and extracellular matrix (based on the GO cellular components database). **(C)** RT-QPCR confirmed silencing of *P3F* in si-GFP compared to control and scramble U23674 cells. **(D, E, F, G, H, I, J)** RT-QPCR also confirmed differential expression of candidate genes in (D, E) cell-to-cell adhesion (*Itga8* and *Cdh4*), (F) cytoskeletal organization (*Rnd2*), and (G, H, I, J) cell-to-extracellular

in siGFP U23674 cells (Fig 7M, upper panel) was reduced in TR100-treated U23674 cells (Fig 7M, lower panel). There were 16.4 ± 3.8 adhesions points per cell in TR100-treated U23674s compared with 35 ± 5.6 adhesion points per cell in DMSO-treated siGFP U23674s ($P$ < 0.01, Fig 7P). In fact, the number of focal adhesion points per cell was similar in TR100-treated siGFP and DMSO-treated scramble U23674 cells (e.g., 16.4 ± 3.8 versus 15 ± 1 adhesion points in TR100-treated siGFP versus DMSO-treated scramble cells, $P$ > 0.9, Fig 7P). Furthermore, more efficient adhesion of siGFP U23674 cells to the surface of the culture dishes was reduced in TR100-treated siGFP U23674 cells ($P$ < 0.0001, Fig 7Q), so that the adhesion capacity of TR100-treated siGFP U23674 cells was similar to that of scramble DMSO-treated U23674 cells ($P$ = 0.03, Fig 7Q). Finally, the higher migration activity seen in transwell migration assays of siGFP U23674 cells (Fig 7U, left panel) was abrogated by exposure to TR100 (Fig 7S–U, right panel). We observed 16 ± 4 migrated cells per well for TR100-treated U23674s compared with 70 ± 16 migrated cells per well for DMSO-treated siGFP cells ($P$ < 0.0001, Fig 7R). Migration of TR100-treated siGFP cells and DMSO-treated scramble cells were similar (e.g., 16 ± 4 versus 11 ± 3 migrated cells per well for TR100-treated siGFP compared with DMSO-treated scramble cells, $P$ > 0.99, Fig 7R).

Similar results were obtained with TR100 treatment of U21459 mouse RMS cells (Fig S14A–R). Again, the organization of the actin cytoskeleton (Fig S14R upper panel) and increased number of focal adhesion points (Fig S14O upper panel, Fig S14S) observed in siGFP U21459 cells was reversed by exposure to TR100 (Fig S14R and O lower panels, Fig S14S). Also, more efficient adhesion to the surface of tissue culture dishes (Fig S14T) and increased transwell migration (Fig S14U–X) seen in siGFP U21459 was reduced by treatment with TR100.

## Discussion

The tumor cell pool in any given cancer is phenotypically and functionally heterogeneous. This heterogeneity arises as a consequence of hierarchical organization, clonal evolution, adaption to microenvironmental, and systemic cues and/or reversible changes in tumor cell properties (27). Cells within the RMS cell pool are known to differ in their expression of cell surface antigens (28), mutational spectrum (11), and degree of tissue-specific differentiation (29). We confirm that low-passage mouse *Myf6Cre+/−,Pax3:Foxo1+/+,p53−/−* RMS cell lines and low-passage human RMS cell cultures contain cells expressing markedly heterogeneous *P3F* levels (17). Within the *Myf6Cre+/−,Pax3:Foxo1+/+,p53−/−* RMS cell pool, a large portion of YFP[high]/P3F[high] cells are in the G2/M phases of the cell cycle, and higher P3F expression correlates with higher proliferation rates. By contrast, YFP[low]/P3F[low] U23674 cells are

mostly in the G0/G1 phases of the cell cycle (17) and reorganize their cytoarchitecture to produce a cellular phenotype prone to adhesion and migration. These differences translate into higher clonal activity and increased tumor-propagating capacity of P3F[low] U23674 cells. Chemical disruption of the actin cytoskeleton, for example, by exposure to the anti-tropomyosin compound TR100 (30), reduced the ability of YFP[low]/P3F[low] mouse RMS cells to adhere and migrate. TR100 and other actin-depolymerizing agents may be of therapeutic value in RMS.

At the single cell level, *P3F* levels fluctuate over time (15). Kikuchi et al reported that *P3F* expression increased in pre-mitotic cells. *P3F* was highly expressed in the G2 cell cycle phase, which correlated with increased *Pax3* promoter activity in G2 and was shown to mediate cell cycle adaptation and survival of cells exposed to genomic stress (15). Our observations suggest that variable *P3F* expression within the RMS cell pool involves transition between phenotypes prone to adhesion and phenotypes predisposed to proliferation. Such adaptive plasticity may provide tumors with a critical advantage during progression through different malignant stages and may allow tumor cells to adapt to environmental challenges. P3F[low] cells may drive metastatic spread at an early stage, which might explain why patients with PAX-translocated RMS frequently harbor micrometastatic disease at first presentation and develop metastases very early (31, 2). Exposure to chemotherapy drugs was shown to change the proportion of P3F[high] and P3F[low] cells, which may contribute to the development of drug resistances.

Interestingly, the melanoma cell pool was recently shown to contain transcriptionally distinct populations of cells, which transition between proliferative and invasive phenotypes to drive melanoma progression (32, 33, 34, 35, 36). Slow-cycling melanoma cells with an invasive phenotype expressed high levels of the receptor tyrosine kinase *AXL* (37), which was one of the most strongly up-regulated genes in YFP[low]/P3F[low] compared with YFP[high]/P3F[high] mouse RMS cells (logFC 2.6, Table S3). Also, Franzetti et al proposed that Ewing sarcomas, a class of bone sarcomas typically expressing the EWS:FLI1 fusion oncoprotein, displayed phenotypic plasticity because of dynamic fluctuations in *EWS:FLI1* expression at the single cell level (38). EWS:FLI1[low] cells were less cohesive and expressed higher levels of actin-binding proteins involved in the assembly of the cytoskeleton compared with EWS:FLI1[high] cells (38). The transition from proliferative to invasive phenotypes was triggered by environmental cues such as extracellular matrix stiffness (33), presence of certain extracellular ligands (e.g., TGFβ) (33), and nutrient/oxygen availability (25). We note that cell densities, glutamine availability, chemical exposures, and certain extracellular matrix components also influenced the proportion of YFP[high]/P3F[high] and YFP[low]/P3F[low] cells within the mouse *Myf6Cre+/−,Pax3:Foxo1+/+,p53−/−* RMS cell pool. Yet, the mechanisms driving

matrix interaction (*Itga5, Thbs3, Efemp1, and Lama5*) in si-GFP compared with control and scramble U23674 cells. **(K)** Assay for transposase-accessible chromatin using sequencing (ATAC-Seq) demonstrated higher ATAC-Seq signals in the *PAX3* promoter region in YFP[high]/P3F[high] cells. **(L)** P3F fusion protein DNA-binding motif represented the most significantly enriched motif in YFP[high]/P3F[high] cells. PCR data were evaluated for statistical significance by ordinary one-way ANOVAs (ns $P$ ≥ 0.05, *$P$ < 0.05, **$P$ < 0.01, ***$P$ < 0.001, ****$P$ < 0.0001). See Figs S5 and S6 for the differential expression of transcripts involved in focal adhesion and in the regulation of the actin cytoskeleton, respectively, in YFP[high]/P3F[high] versus YFP[low]/P3F[low] cells. See Table S3 for a list of genes differentially regulated genes in YFP[high]/P3F[high] versus YFP[low]/P3F[low] cells (logFC < −1 or >1, false discovery rate [FDR] < 0.01); see Table S4 for pathways enriched in genes differentially regulated (logFC < −1 or >1, FDR < 0.01) in YFP[high]/P3F[high] versus YFP[low]/P3F[low] cells (FDR < 0.05); see Tables S5 and S6 for genes with stronger ATAC-Seq signals in YFP[low]/P3F[low] and YFP[high]/P3F[high] U23674 cells; see Table S7 for pathways enriched among genes with stronger ATAC-Seq signals in YFP[low]/P3F[low] cells.

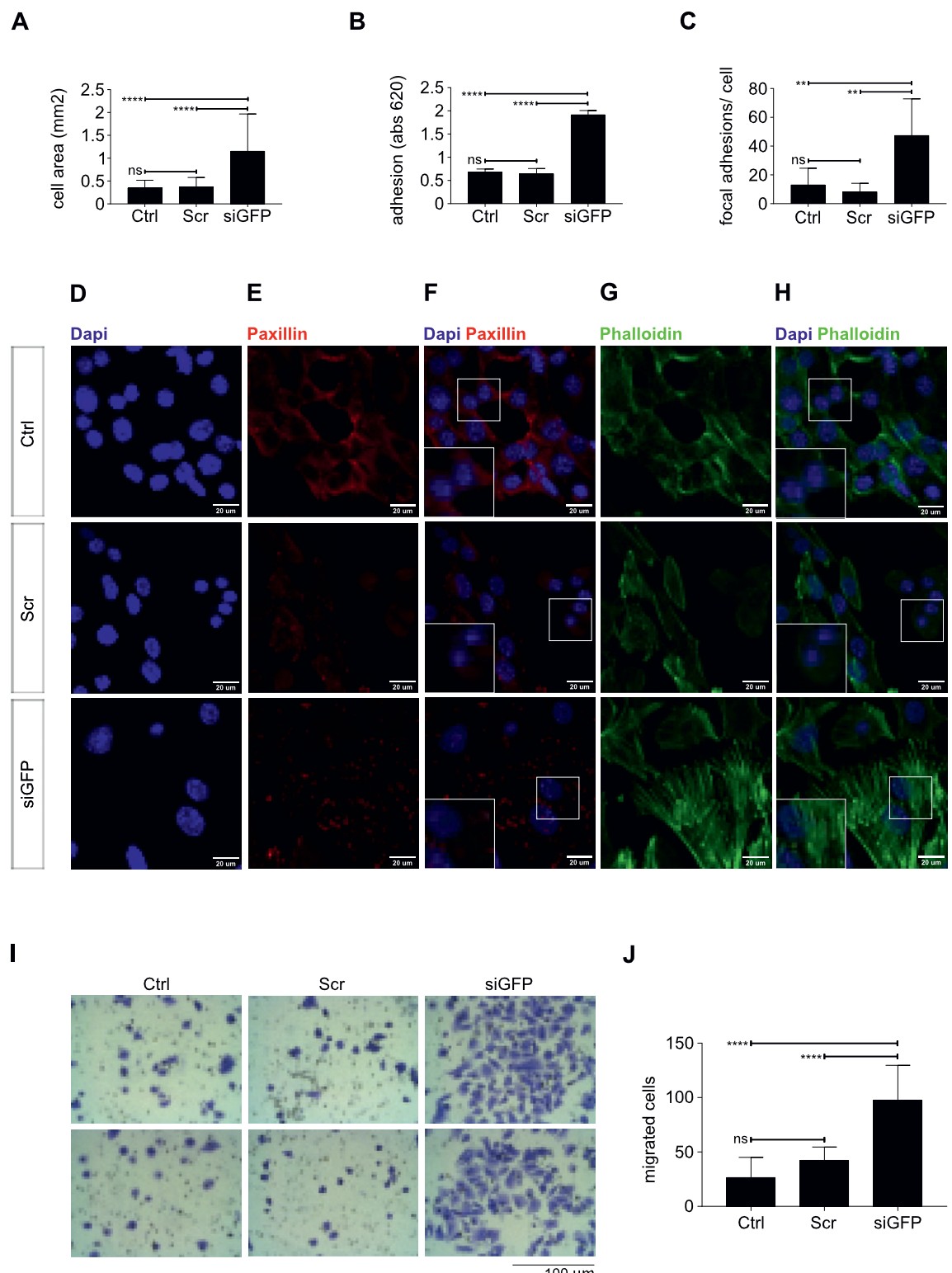

**Figure 6. Lower P3F dose in mouse U23674 RMS cells changed cytoarchitecture, adhesion and migration.**
**(A)** The cell surface area of si-GFP U23674 cells was higher than that of control and scramble U23674 cells 2 h after plating. **(B)** Also, the number of adherent si-GFP cells was higher than that of control and scramble cells, as evidenced by Toluidine Blue staining 2 h after plating. **(C, D, E, F)** Si-GFP cells (F, bottom panel) contained more focal adhesion points per cell compared to control and scramble cells (F, upper and middle panels), as visualized by Paxillin immunocytochemical staining. **(G, H)** Phalloidin staining revealed re-organization of the actin cytoskeleton with formation of robust stress fibers in si-GFP cells (H, bottom panel) compared with control and scramble cells (H, upper and middle panels). **(I, J)** Si-GFP cells (I, far right panels) exhibited higher migration capacity than control and scramble cells (I, far left and middle panels).

fluctuating *P3F* expression in mouse RMS remain unclear. It is important to note that the endogenous *Pax3* promoter, which drives *P3F* expression in *Myf6Cre+/−,Pax3:Foxo1+/+,p53−/−* mouse RMS cells (14), displayed higher ATAQ-Seq signals in YFP^high/P3F^high U23674 cells. We speculate that changes in *P3F* expression may be a consequence of core regulatory circuits that establish and maintain cellular properties through their extended regulatory networks (39).

P3F^low RMS cells are mostly in the G0/G1 phases of the cell cycle, prone to adhesion and with higher tumor-propagating potential than P3F^high cells. It is conceivable that they represent a stem-like cellular state. This is interesting, as increasing evidence supports that extracellular matrix proteins provide a physical and biochemical niche to promote stem cell survival by establishing anchorage/homing sites that serve as a reservoir for external factors, allow for formation of focal adhesions and mediate activation of mechanotransduction pathways (40). The ability to bind to extracellular matrix components was previously linked to clonogenic activity and tumor-forming capacity in sarcoma cells. Buchstaller et al demonstrated that laminin-negative malignant peripheral nerve sheet tumor (MPNST) cells displayed lower tumor-forming capacity than in laminin-positive MPNST cells. Lower tumor-forming capacity of laminin-negative MPNST cells was augmented by coinjecting cells with laminin/Matrigel into mouse recipient animals (41).

Cell-to-cell heterogeneity in *P3F* expression was present within the mouse *Myf6Cre,Pax3:Foxo1,p53* RMS cell pool, but also in human primary RMS cell cultures and RMS cell lines. However, one important limitation of our studies is that we were unable to reproduce a correlation between lower P3F expression and higher capacity to adhere to a variety of different surface matrices in human RMS cell lines Rh30, Rh41, or Rh5, which could be due to fundamental differences in mouse and human tumors or, more likely, due to artefacts introduced by long-term passage of the human cell lines in vitro. Our subsequent attempts to silence *P3F* expression in low-passage human primary RMS cell cultures were hampered by substantial toxicities and poor survival of siP3F-transfected cells. *P3F* silencing in CF1 primary human RMS cells was not associated with changes in cell adhesion to plastic surfaces. Of note, melanoma studies also indicated that ectopic expression of the melanoma transcriptional master regulator microphthalmia-associated transcription factor (MITF) did not necessarily induce phenotype switching, possibly because epigenetic modifications were necessary for MITF to drive the transition between phenotypes (32). Further single-cell analyses will be needed to clarify differences in gene expression signatures linked to distinct phenotypes within the RMS cell pool.

Taken together, our studies highlight variable P3F expression at the cellular level in human primary RMS cell cultures and *Myf6Cre+/−,Pax3:Foxo1+/+,p53−/−* mouse RMS tumors, and we demonstrate the functional consequences of this heterogeneity with respect to tumorigenic and invasive potential. Importantly, higher

proportions of cells in the G0/G1 phase of the cell cycle among YFP^low/P3F^high cells may contribute to their higher tumor-propagating and clonogenic capacity. Because of its central role in RMS malignancy, the P3F fusion oncogene has generally been considered an ideal target to selectively attack tumor cells. Yet, our data clearly indicate that eliminating P3F^high cells only by targeting the fusion oncogene may not cure the disease. This is supported by published observations in genetically engineered mouse RMS tumors, which regressed after withdrawal of inducible *P3F* followed by rapid recurrence without re-activation of *P3F* (42). It will be important to understand the mechanisms that direct fluctuations in P3F expression at the cellular level, to design treatment regimens, that might be able to overcome plasticity, for example, by using metronomic therapies that avoid adaptation, prevent development of resistance and set the stage for each other.

# Materials and Methods

### Mice

*NOD/CB17-Prkdc^scid/J* (NOD.SCID) mice were bred and maintained at the Center for Experimental Models and Transgenic Service (CEMT) Freiburg. All animal experiments were approved by the Regierungspräsidium Freiburg (G-16/136).

### Cell lines

Mouse U23674 and U21459 RMS cell lines were established in mouse sarcomas, which arose spontaneously in *Myf6Cre,Pax3:Foxo1,p53* mice (Table S1). In these mice, expression of *Myf6-Cre* converted the two normally functioning *Pax3* alleles into conditional *P3F* knock-in alleles by fusing exons 1–7 of *Pax3* to a 9.3-kb 3′ genomic region of *Foxo1*. *P3F* was linked to an *eYFP* fluorescent marker gene, which was expressed as a second cistron downstream from an IRES. The mice also carried conditional *Tp53* knockouts on both alleles. For the experiments reported here, U23674 cells were used at passage 13–24, and U21459 at passage 7–24. U23674 and U21459 cells were grown in DMEM (41965-039; Gibco), supplemented with 10% FBS (F7524; Sigma-Aldrich), and 1% Penicillin/Streptomycin (PS, 15140-122; Gibco).

SJRHB013759_X1 primary RMS cell cultures were established from a recurrent inguinal *P3F* fusion-positive RMS tumor diagnosed in a 19-yr-old male, IC-pPDX 35 primary RMS cell cultures from a recurrent *P3F* fusion-positive RMS tumor diagnosed in a 13-yr-old male, RMSZH003 from a recurrent pelvic *P3F* fusion-positive RMS tumor diagnosed in a 3-yr-old female and CF1 from a 1.8-yr-old boy with disseminated disease (43). For the experiments reported here, SJRHB013759_X1 cells were used at passage 5–9, IC-pPDX 35 cells at passage 3–9, RMSZH003 cells at passage 4–7 and CF1 at passage 7–15

---

Data were evaluated for statistical significance by ordinary one-way ANOVA statistical test (ns $P \geq 0.05$, *$P < 0.05$, **$P < 0.01$, ***$P < 0.001$, ****$P < 0.0001$). Experiments were replicated three times. See Fig S9 for the effects of *P3F* silencing on the cytoarchitecture, adhesion, and migration of mouse U21459 cells. See Fig S10 for the effects of *P3F* silencing on the expression of candidate genes involved in cell-to-cell adhesion, cytoskeletal organization, and cell-to-extracellular matrix interaction in mouse U21459 cells. See Fig S11 for the effects of *P3F* silencing on the adhesion capacities of human Rh5, Rh30, and Rh41 cells. See Fig S12 for the effects of *P3F* silencing on the adhesion capacities of CF1 human patient-derived RMS cell cultures.

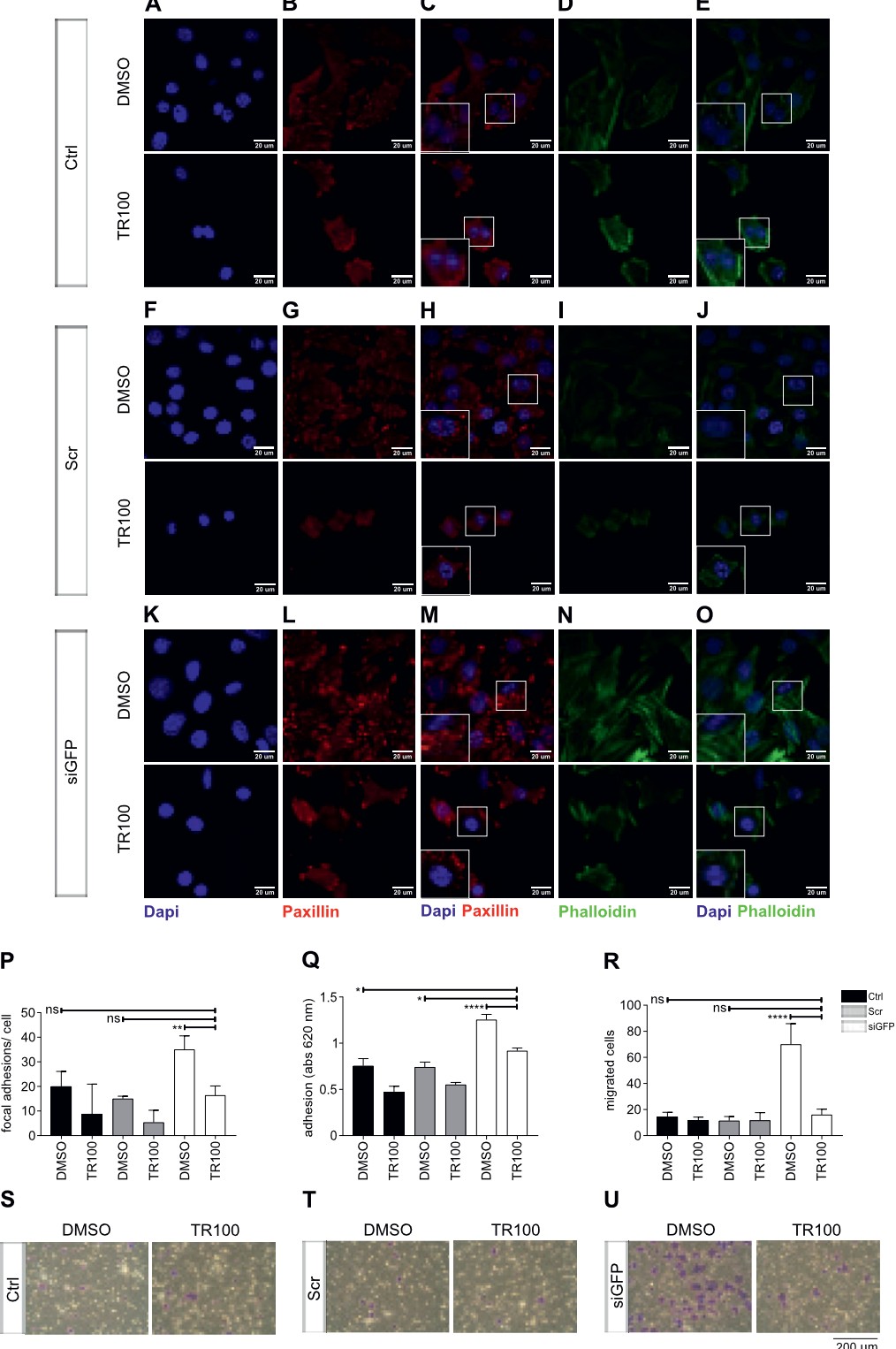

**Figure 7. Chemical disruption of the actin cytoskeleton in U23674 mouse RMS cells reversed the effects of lower P3F dose on cell adhesion and migration.**
**(A, B, C, D, E, F, G, H, I, J, K, L, M, N, O)** Visualization of the actin cytoskeleton and focal adhesion points per cell by phalloidin and paxillin staining of U23674 cells treated with the anti-tropomyosin compound TR100 or carrier only (DMSO). **(O)** TR100 disrupted increased stretching of the actin cytoskeleton in siGFP cells. **(M)** TR100 also reversed the higher number of focal adhesion points per cell in siGFP cells. **(P)** Quantification of focal adhesion points per cell confirmed that TR100 reduced the higher number of adhesion points observed in DMSO-treated siGFP cells to numbers than those observed in control and scramble cells treated with DMSO. **(Q, R, S, T, U)** Exposure to TR100 also abrogated (Q) more efficient adhesion and (R, S, T, U) higher migration capacity of DMSO-treated siGFP cells to levels similar to those observed in control and scramble cells treated with DMSO. Data were evaluated for statistical significance by ordinary one-way ANOVAs (ns $P \geq$ 0.05, *$P <$ 0.05, **$P <$ 0.01, ***$P <$ 0.001, ****$P <$ 0.0001). Experiments were replicated three times. See Fig S13 for the effects of TR100 treatment on organization of the actin cytoskeleton, adhesion and migration in mouse U21459 cells.

(Table S1). SJRHB013759_X1, IC-pPDX 35, and RMSZH003 were cultured in Neurobasal Medium (10888022; Gibco) supplemented with 1% penicillin/streptomycin (15140-122; Gibco), 1× Glutamax (35050; Gibco), 2× B27 (17504044; Life Technologies), 20 ng/ml bFGF (AF-100-

18B; Peprotech), and 20 ng/ml EGF (AF-100-15; Peprotech). CF1 were grown in RPMI supplemented with 10% FBS and 1% PS.

Mouse C2C12 cells and human RD cells (used as YFP^neg control cells), mouse *Kras(G12v)*; *P16p19^null* mouse RMS cells (used as YFP^pos

control cells [44]); and human HEK293T cells were grown in DMEM supplemented with 10% FBS and 1% PS. Rh5, Rh30 and Rh41 human RMS cells (Table S1) were grown in DMEM supplemented with 20% FBS and 1% PS.

Short tandem repeat analyses of human (Table S8) and mouse (Table S9) cell lines used in this study was performed by Eurofins.

## FACS

Cells were suspended in HBSS with 2% FBS. Antibody staining was performed for 20 min on ice. Before FACS sorting, cells were labeled with 7AAD (559925; BD Biosciences) to exclude dead cells. Cells were sorted twice using a MoFlo Astrios flow cytometer. Purity checks were performed to confirm that the sorted YFP$^{pos}$ and YFP$^{neg}$ cell subsets had a purity of >98% using a YFP expression threshold determined by the background fluorescence of YFP$^{neg}$ C2C12 cells.

## Cell cycle analysis

U23674 cells were stained with Hoechst 33342 (H1339; Invitrogen) at a final concentration of 5 μg/ml for 45 min at 37°. Cells were then centrifuged at 300$g$ for 5 min and resuspended in HBSS supplemented with 2% FBS. FACS analysis was performed on LSRFortessa from Becton Dickinson.

## Clonal assays

Twenty U23674 cells per well were sorted into 96-well plates and allowed to expand into clones. Formation of clones was evaluated after 6–11 d using a microscope (Axiovert 40C Microscope; Carl Zeiss). Clones were harvested 11 d after plating and subjected to RNA isolation or flow cytometry analysis.

## Sarcoma transplantation

Sarcoma cells were sorted, reconstituted in HBSS with 2% FBS, and injected at defined numbers into the gastrocnemius muscles of 1–3-mo-old, anesthetized NOD.SCID mice as previously described (44). Recipient tissue was preinjured 24 h before cell injection with 25 μl of a 0.03 mg/ml solution of cardiotoxin (from *Naja naja mosambica*; Sigma-Aldrich). Mice were followed up for up to 4 mo after transplantation. The extremity muscles of mice that did not develop palpable tumors were dissected 4 mo after transplantation to exclude tumors.

Tumor-bearing mice were euthanized, and tumors were harvested. Tumor tissue was digested in DMEM supplemented with 0.2% collagenase type II (17101-015; Gibco) and 0.05% Dispase (17105-041; Gibco) for 90 min at 37° in a shaking water bath and mechanically dissociated as described before (22). Red blood cells were removed using ammonium-chloride-potassium (ACK) Lysing Buffer (A1049201; Thermo Fisher Scientific) for 3 min on ice.

## Transcriptional profiling and pathway analysis

Total RNA was isolated using TRIzol Reagent (15596018; Ambion) and quantified using a 2.0 Qubit fluorometer (Invitrogen). RNA integrity was confirmed using an Agilent 2100 Bioanalyzer. Kit, Clariom S

Assay, and mouse array (902931; Affymetrix) was performed according to the manufacturer's instructions. CEL files were processed with the Oligo R package (45) and intensity were RMA normalized. A linear-based model (46) was used to identify differentially regulated genes in YFP$^{high}$/P3F$^{high}$ versus YFP$^{low}$/P3F$^{low}$. Regulated genes with a fold-change (logFC) < −1 or >1 and a FDR < 0.01 were selected for gene set analysis using Fisher's exact test. Databases were downloaded from MSigDB (47) Consensus pathDB (48). Raw data are accessible on gene expression omnibus (GEO) using the following GSE ID: GSE153894.

## Transposase-accessible chromatin using sequencing (ATAC-Seq)

ATAC-Seq was performed on mouse U23674 YFP$^{high}$/P3F$^{high}$ and YFP$^{low}$/P3F$^{low}$ cells as previously described (26). Briefly, 23k cells were collected and centrifuged at 500$g$ for 5 min at 4°C. The supernatant was discarded without disturbing the cell pellets, which were resuspended in 50 μl of cell lysis buffer (Tris–HCl, pH 7.4, 10 mM NaCl, 3 mM MgCl$_2$, and 0.1% NP40) and centrifuged at 500$g$ for 5 min at 4°C. The supernatant was removed carefully and resuspended in freshly prepared Tn5 reaction buffer (12.5 μl 1X TD buffer [Illumina], 1 μl Tn5 transposase, and 11.5 μl nuclease-free water). The transposition reaction was incubated at 37°C for 30 min and then purified using the Zymo ChIP clean up kit. Transposed DNA was subjected to five cycles of indexed PCR amplification using New England Biolabs Next Ultra II Q5 Master Mix and custom-indexed primers. Quantitative RT–PCR was performed to determine the optimal number of PCR cycles for linear amplification without saturating ATAC-Seq libraries. ATAC-Seq libraries were then quantified using Qubit (Invitrogen), normalized, pooled, and sequenced on an Illumina HiSeq 4000 sequencer (paired-end 125 bp, on average 100 M reads per library). FASTQ files were processed by trimming Illumina adapters and Tn5 sequences with trimmomatic before alignment to the mouse genome (build mm10) using bowtie2. Duplicate and mitochondrial reads were removed. The HOMER pipeline was used to determine transcription factor motifs, which were enriched in specific cell types (49). Gene Set Enrichment Analysis was performed as previously described (47, 50). Raw data are accessible on GEO using the following GSE ID: GSE154452.

## Si-RNA silencing and retroviral transduction

SiRNA-silencing was performed using Lipofectamine RNAiMax Transfection Reagent (13778030; Invitrogen) according to the manufacturer's instructions. SiRNAs were obtained from Dharmacon (ON-TARGET plus nontargeting control pool for the scramble and si-GFP). pMSCV-Flag-IRES-GFP retrovirus was produced in HEK293T cells, cotransfected with pMSCV-Flag-IRES-GFP (29.3 μg), pCMV-Gag-Pol (9.75 μg), and pMD2.VSV.G (4.9 μg). Retrovirus-containing supernatant was concentrated by ultracentrifugation at 19,500 rpm (rMAX 161, rMIN 75.3, rAV 118.2mm) using a Sorvall WX Ultra Series 80 centrifuge (Thermo Fisher Scientific) for 3 h at 4°C.

## Annexin V staining

Apoptosis was evaluated by Annexin V-APC staining (550474; BD Biosciences) according to the manufacturer's protocol using a FACS canto flow cytometer. 7AAD was used for viability staining. U23674

and Rh30 cells treated with etoposide at a concentration of 50 $\mu$M were used as positive controls.

## Western blotting

Cell pellets were lysed using cell lysis buffer (9803S; New England Biolabs) supplemented with protease/phosphatase inhibitor cocktail (5872S; Cell Signaling Technology). Proteins were resolved on SDS-polyacrylammide gels and blotted onto Immuno-Blot polyvinylidene difluoride (PVDF) membranes (1620177; Bio-Rad), which were blocked with PBST 3% non-fat dry milk, incubated with primary antibodies overnight at 4°C, washed, and hybridized for 1 h at room temperature using goat anti-mouse/rabbit immunoglobulin G (IgG) (H + L)–HRP Conjugate (1706516/15; Bio-Rad) depending on the origins of the primary antibody. Detection was performed using the ECL Select Western Blotting Detecting Reagent (RPN2235; Amersham). The following antibodies were used: anti-$\beta$ actin (1:50,000, AC15; Sigma-Aldrich), anti-Pax3:Foxo1 (1:500, C29H4; Cell Signaling Technology), anti-PARP (1:1,000, 9542; Cell Signaling Technology), and anti-Cleaved Caspase 3 (1:500, 9664; Cell Signaling Technology).

## Real-time PCR

Total RNA was isolated using TRIzol Reagent (15596018; Ambion) according to the manufacturer's instructions and quantified using a NanoDrop Spectrophotometer (Thermo Fisher Scientific). RNA was reverse-transcribed using Superscript III First Strand (18080051; Invitrogen). Real-time PCR was performed using SybrGreen (4309155; Thermo Fisher Scientific). The relative expression of each gene was defined from the threshold cycle (Ct), and relative expression levels were calculated by using the 2-DDCt method. Mouse *Gapdh* or *Actin* was used as housekeeping genes. Primer sequences are listed in Table S10.

## Single-cell reverse transcriptase droplet digital PCR (RT-ddPCR)

One RMS cell per well was sorted in 96-well plates containing 4.5 $\mu$l Single-Cell Lysis Buffer and 0.5 $\mu$l Single-Cell DNase I (Single-Cell Lysis Kit, 4458235; Ambion) using a Moflo Astrios. The reaction was stopped by adding 0.5 $\mu$l of single-cell stop solution. cDNA synthesis was performed after adding 2.5 $\mu$l of RT reaction mix (iScript Advanced cDNA Synthesis kit, 1725038; Bio-Rad) for a total volume of 10 $\mu$l. Droplet digital PCR amplification of *Pax3:Foxo1* (FAM) and *Gapdh* (HEX) was performed in a final volume of 25 $\mu$l by adding the ddPCR Supermix for Probes No UDP (Bio-Rad), the *FAM* and *HEX* probes to the cDNA mix. Droplets were generated using the QX100 Droplet Generator (Bio-Rad) with 70 $\mu$l Droplet Generation Oil (Bio-Rad) and 40 $\mu$l of the resulting water-in-oil droplet emulsion was then thermocycled at 95°C for 10 min, followed by 40 cycles of 95°C for 30 s and 61.3°C for 1 min. Final enzyme deactivation took place at 98° for 10 min. Individual droplets were analyzed using the Q100 Droplet Reader and Quantasoft Software. The following probe sequences were used: 5′-/56-FAM/CATTGGCAA/ZEN/TGGCCTCTCACCTCAGAA/3IABkFQ/-3′ (*P3F* FAM probe), 5′-/5HEX/ACCACAGTC/ZEN/CATGCCATCACTGCCACC/3IABkFQ/-3′ (*GAPDH* HEX probe).

## Immunocytochemistry

Cells were fixed using 4% PFA, permeabilized with 0.2% Triton X-100, blocked with Vector M.O.M. Immunodetection Kit (BMK-2202) containing 10% goat serum, and incubated with Alexa Fluor 488–conjugated anti-phalloidin (1:1,000, A12379; Thermo Fisher Scientific) at room temperature for 1 h, Alexa Fluor 594-conjugated anti-Phalloidin (1:100, ab176757; Abcam) at room temperature for 1 h or anti-Paxillin (1:100, 610051; BD Biosciences) at 4°C overnight. Alexa Fluor 594 goat anti-mouse IgG (1:200) was used for secondary antibody staining at room temperature for 1 h. Antibodies were diluted in Vector M.O.M. Immunodetection Kit (BMK-2202) containing 10% goat serum. Nuclei were stained with Dapi. Images were obtained using a Zeiss LSM 710 confocal microscope.

## Adhesion assays

Cells were seeded at 100,000 cells per well using non-coated 24-well plates and allowed to adhere to the plate for 2 h at 37° before removal of the supernatant and non-adherent cells. To quantify adhesion, cells were fixed with 4% PFA for 30 min, washed with PBS, stained with Toluidine Blue for 1 h, air dried overnight, and dissolved in 2% SDS solution. Optical density was measured at 620 nm. To evaluate cell surface areas, adherent cells were imaged using a HBO 100 AXIO microscope (Carl Zeiss), and cell surface areas were measured for 15 representative fields using ImageJ.

For the human Rh5, Rh30, and Rh41 sarcoma cell lines, adhesion assays were carried out using the ECM Cell Adhesion Array Kit (ECM540; Millipore) according to the manufacturer's instructions. Briefly, cells were seeded at 20,000 cells per well, allowed to adhere for 2 h at 37° and stained according to the manufacturer's instructions. The absorbance was then measured at 560 nm. For each well, the ratio between the absorbances obtained for cells grown on coated and non-coated surfaces was calculated.

## Migration assay

Cells were seeded at 300,000 cells per well in serum-free medium using cell culture inserts (353093; Falcon) in six-well plates. Cells were allowed to migrate for 8 h at 37°. Migrated cells were then stained with crystal violet for 30 min. Excess crystal violet was removed, and the inserts were air dried overnight. Cells were imaged using a HBO 100 AXIO microscope (Carl Zeiss).

## Drug exposures

U23674 cells were pretreated with 10 $\mu$M TR100 (SML1065; Sigma-Aldrich) and U21459 cells with 15 $\mu$M TR100 for 12 h and then seeded for paxillin/phalloidin immunocytochemistry staining, adhesion, and migration assays as described above. Control cells were treated with vehicle (DMSO) only.

To evaluate the drug effects on the proportion of YFP[high]/P3F[high] and YFP[low]/P3F[low] U23674 RMS cells, U23674 cells were seeded at 30,000 cells/well in 96-well plates and then treated with 2.5 nM dactinomycin (Recordati) for 48 h, 5 nM vincristine (Tewa Ratiopharm) for 48 h, or 10 $\mu$M TR100 for 12 h. Control cells were treated with vehicle only. The percentage of YFP[high]/P3F[high] U23674 RMS cells was measured by FACS.

## Environment effects on the proportion of YFP$^{high}$/P3F$^{high}$ U23674 RMS cells

To evaluate nutrient effects, cells were seeded at 50,000 cells/well in six-well plates (flat bottom) in medium containing 0.5 mM glutamine and 25 nM glucose. After 24 h, the medium was replaced with medium containing increasing concentrations of glutamine (0.05, 0.5, and 4 mM) and 25 nM glucose. Alternatively, the medium was replaced with a medium containing increasing concentrations of glucose (2.8, 25 nM) and 0.5 mM glutamine. To evaluate cell density effects, cells were seeded at 10,000 cells/well in flat bottom and triangle bottom 96-well plates in DMEM. Alternatively, cells were plated in flat bottom 96-well plates in DMEM at increasing cell densities as indicated. Finally, to evaluate matrix effects, surfaces were coated with laminin 50 µg/ml (L2020; Sigma-Aldrich), Matrigel (354234; Corning), and fibronectin 5 µg/ml (FC010; Millipore). The percentage of YFPhigh/P3Fhigh U23674 RMS cells was evaluated by FACS 72 h after changing the medium or 48 h after seeding.

### Statistics

Differences in % YFP expression, candidate gene expression by RT-QPCR, proportion of Annexin V-/7AAD-living cells, proportion of Annexin V+/7AAD-apoptotic cells, proportion of Annexin V+/7AAD+ necrotic cells, cell surface areas, cell adhesion, number of focal adhesions/cell as evidenced by paxillin staining, and cell migration were tested for significance using one-way ANOVAs with Bonferroni's multiple comparisons. Environmental differences in the percentage YFP$^{high}$/P3F$^{high}$ and YFP$^{low}$/P3F$^{low}$ cells and changes in the adhesion capacity of human RMS cells were tested for significance using two-tailed $t$ test for unpaired data. Differences in cell growth were evaluated by nonlinear regression analysis. Differences in clonal activity were tested for significance using a chi square test. Differences in tumor-propagating capacity were evaluated statistically using log-rank (Mantel–Cox) tests. Limiting dilution analyses were performed based on Bonnefoix et al ([51]) using the limdil function of the StatMod package (author GK Smyth, http://bioinf.wehi.edu.au/software/limdil/). For single-cell digital droplet PCR data, *P3F* expression was normalized based on *GAPDH* expression, and cell-to-cell variability was illustrated using the interquartile range of the normalized expression median.

## Data Availability

The datasets produced in this study are available in the following databases: Raw RNA-Seq data are accessible on GEO using the following GSE ID: GSE153894. Raw ATAQ-Seq data are accessible on GEO using the following GSE ID: GSE154452.

## Supplementary Information

## Acknowledgements

We thank J Bodinek-Wersing at the Lighthouse Core Facility of the Albert Ludwigs Universität Freiburg for excellent flow cytometry support. We are grateful to D Pfeifer in the genomics laboratory at University Medical Center Freiburg for help with microarray processing and to S Burdach, J Rohr, C Flotho, and M Erlacher for helpful comments on the data. This work was funded by the Deutsche Forschungsgemeinschaft (DFG [HE 7673/1-1]; to S Hettmer). S Hettmer was supported by the Berta-Ottenstein-Program for Advanced Clinician Scientists, Faculty of Medicine, University of Freiburg. AJ Wagers was supported by an National Institutes of Health (NIH) Pioneer Award (DP1 OD025432) and Joslin Diabetes Center funds. The article processing charge was funded by the Baden-Wuerttemberg Ministry of Science, Research and Art and the University of Freiburg in the funding programme Open Access Publishing

### Author Contribution

C Regina: formal analysis, investigation, visualization, methodology, and writing—original draft, review, and editing.
E Hamed: formal analysis, investigation, and writing—review and editing.
G Andrieux: formal analysis, investigation, methodology, and writing—review and editing.
S Angenendt: investigation and writing—review and editing.
M Schneider: investigation.
M Ku: formal analysis, investigation, methodology, and writing—review and editing.
M Follo: formal analysis, investigation, methodology, and writing—review and editing.
M Wachtel: resources, formal analysis, and writing—review and editing.
E Ke: formal analysis and writing—review and editing.
K Kikuchi: investigation and writing—review and editing.
AG Henssen: investigation and writing—review and editing.
BW Schäfer: resources, formal analysis, and writing—review and editing.
M Boerries: formal analysis, investigation, methodology, and writing—review and editing.
AJ Wagers: conceptualization, resources, formal analysis, and writing—review and editing.
C Keller: conceptualization, resources, formal analysis, and writing—review and editing
S Hettmer: conceptualization, resources, data curation, formal analysis, supervision, funding acquisition, investigation, visualization, methodology, project administration, and writing—original draft, review, and editing

### Conflict of Interest Statement

The authors declare that they have no conflict of interest.

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
