## [Reviewer comments · Life Science Alliance]

Life Science Alliance

Negative correlation of single-cell PAX3:FOXO1 expression with tumorigenicity in rhabdomyosarcoma.

Carla Regina, Ebrahim Hamed, Geoffroy Andrieux, Sina Angenendt, Michaela Schneider, Manching Ku, Marie Follo, Marco Wachtel, Euge Ke, Ken Kikuchi, Anton Henssen, Beat Schaefer, Melanie Bőrries, Amy Wagers, Charles Keller, and Simone Hettmer

DOI: <https://doi.org/10.26508/lsa.202001002>

Corresponding author(s): Simone Hettmer, University Medical Center Freiburg, Pediatric Hematology and Oncology

Review Timeline:

Submission Date:	2020-12-22
Editorial Decision:	2021-01-29
Revision Received:	2021-05-18
Editorial Decision:	2021-06-11
Revision Received:	2021-06-16
Accepted:	2021-06-17

Transaction Report:

January 29, 2021

Re: Life Science Alliance manuscript #LSA-2020-01002-T

Dr. Simone Hettmer
University Medical Center Freiburg, Pediatric Hematology and Oncology
Pediatric Hematology and Oncology
Mathildenstrasse 1
Freiburg 79106
Germany

Dear Dr. Hettmer,

Thank you for submitting your manuscript entitled "Negative correlation of single-cell PAX3:FOXO1 expression with tumorigenicity in rhabdomyosarcoma." to Life Science Alliance. The manuscript was assessed by expert reviewers, whose comments are appended to this letter.

As you will note from reviewers' comments below, both reviewers are enthusiastic about these findings, but do raise some concerns that should be addressed prior to further consideration of this study at Life Science Alliance (LSA). We, thus, invite you to submit a revised manuscript that addresses all the points raised by Rev 1, and pt #2 and minor concerns raised by Rev 2. The descriptive nature of the study and the lack of in vivo data would not preclude further consideration of the manuscript at LSA.

Thank you for this interesting contribution to Life Science Alliance. We are looking forward to receiving your revised manuscript.

Sincerely,

Shachi Bhatt, Ph.D.
Executive Editor
Life Science Alliance
<https://www.lsa-journal.org/>
Tweet @SciBhatt @LSAJournal

- A letter addressing the reviewers' comments point by point.
- An editable version of the final text (.DOC or .DOCX) is needed for copyediting (no PDFs).
- High-resolution figure, supplementary figure and video files uploaded as individual files: See our detailed guidelines for preparing your production-ready images, <https://www.life-science-alliance.org/authors>
- Summary blurb (enter in submission system): A short text summarizing in a single sentence the study (max. 200 characters including spaces). This text is used in conjunction with the titles of papers, hence should be informative and complementary to the title and running title. It should describe the context and significance of the findings for a general readership; it should be written in the present tense and refer to the work in the third person. Author names should not be mentioned.

B. MANUSCRIPT ORGANIZATION AND FORMATTING:

Reviewer #1 (Comments to the Authors (Required)):

Summary

Rhabdomyosarcoma (RMS) is a soft tissue sarcoma in children and teens. There are two major subtypes. A Fusion negative subtype where RAS mutations are generally present in tumors with poor outcome and Fusion positive subtype is driven by a fusion oncogene containing either PAX3 or PAX7 fused to the FOXO1. While in FN-RMS tumors cell heterogeneity and the presence of cancer

stem cells has been previously defined; however, in FP-RMS differences in tumorigenicity between sub-populations or tumors cell heterogeneity has not been clearly delineated and remains an open question with implications in how this tumor subtype propagates and responds to therapy.

Work by Keller and colleagues using a mouse model of FP-RMS in which the PAX3FOXO fusion oncogene is expressed along with YFP from the endogenous PAX3 promoter in Trp53^{-/-} animals. These studies have shown that there is heterogeneity in the expression of the fusion oncogene which is expressed at higher levels in the G2 phase of the cell cycle. This study by Regina et al., which uses low passage mouse FP-RMS cells generated from the murine model adds significantly to defining effects of differential levels of the PAX3FOXO oncogene on the ability to propagate tumor in vitro and in vivo. Both the in vitro clonal analyses and in the in vivo mouse xenograft experiments clearly show that the PAX3FOXO1 low cells have a significantly higher tumorigenic potential compared to PAX3FOXO Hi cells. Importantly, when high and low cells are sorted and plated they give rise to clonal growth and tumors with mixed populations similar to the unsorted parental cell line. The authors also confirm that there is heterogeneity in the fusion oncogene in human FP-RMS cell lines and low passage Patient derived xenografts. Additionally, the authors show that the proportion of cells expression PAX3FOXO1 fusion levels in the U23674 cell line can change in response to changes in the medium, the substratum, density and perturbation with vincristine or dactinomycin. Overall, this study is able to assess an important question in the field about the effect of heterogeneity in PAX3FOXO1 expression and effects on tumorigenicity. However, more broadly this study also highlights some of the challenges in treating tumors with heterogeneous tumor populations that can respond differently to the current standard of care.

Comments

1. While a single murine derived FP-RMS cell line U23674 used in all the analyses some of the major effects on heterogeneity in PAX3FOXO are not observed in the human cell lines; while this difference may be due to the human FP-RMS changing from being in culture for many years, it could also be that the U23674 cell line is an outlier. Can the authors perform similar in vitro assays in Figure 3 with the other mouse cell line U21459 i.e. growth in a) high glutamine b) laminin or matrigel c) higher densities of cells d) treatment with vincristine and dactinomycin.

2. There is a paradox associated with the data in Figure 3 and Figure 4. PAX3FOXO YFP high cells are more proliferative while PAX3FOXO low/negative cells are slow cycling but more clonogenic and tumorigenic. However, it appears that when the cells are stressed or allowed to overgrow which is usually associated with slower growth there is an increase in PAX3FOXO YFP high cells. Thus this data can also suggest that when exposed to stress high PAX3FOXO protects cells from apoptosis by inducing a G2M block in cycle rather than the cells actually proliferating.

Can the authors perform Cell cycle along with apoptosis analyses comparing PAX3FOXO YFP high to low cells in U23674 cell grown in a) high glutamine b) higher densities of cells c) treatment with vincristine and dactinomycin. This will help determine if the high PAX3FOXO state is the one that is able to survive treatment while the low PAX3FOXO state then allows invasion migration and tumorigenic seeding.

3. The data provided in figure 4 A and B do not match the text. In the plots provided it appears that PAX3FOXO YFP hi cells have significantly lower Necrotic/late apoptotic cells compared to unsorted or PAX3FOXO YFP low cells. Can the authors comment or provide new plots.

4. In Figure 4E-G can the authors better explain the rationale for switching from experiments with

sorted YFP hi and YFP low cells to experiments with knocking down PAX3FOXO with siRNA. For example, are these cells mostly in the G0/G1 phase of the cell cycle, and more sensitive to vincristine treatment and more tumorigenic/clonogenic?

Reviewer #2 (Comments to the Authors (Required)):

Life Science Alliance: Negative correlation of single-cell PAX3:FOXO1 expression with tumorigenicity in rhabdomyosarcoma (ID# LSA-2020-01002-T)

This manuscript by Regina et al describes differing phenotypes between PAX3-FOXO1 (PF3) RMS cells expressing high levels and low levels of PF3. PF3-RMS is a deadly disease and an important clinical problem. The authors describe that RMS cells with low levels of PF3 produce a cellular phenotype characterized by more efficient adhesion and migration, which correlate with higher tumor propagating frequencies. The authors then go on to show that exposure to agents that disrupt the cytoskeleton reverse enhanced migration and adhesion of RMS cells. The authors therein conclude that heterogeneous expression of PF3 at the single cell level may provide a critical advantage during tumor progression.

While this manuscript is well communicated and the authors have provided data consistent with their conclusions, the manuscript in its present form is unfortunately rather descriptive, and thus lacks true mechanistic insight and clinical translation to human RMS - and is not yet ready for publication in LSA. The reasons for this assessment from this reviewer include the following:

1) Unfortunately, one major weakness of this manuscript is the lack of in vivo data. Other than an early xenograft study, the authors do not provide any in vivo data to augment their findings in the latter two-thirds of the manuscript. For example, different matrices that were used in the in vitro studies (or equivalents) and/or the therapeutics used on cells could have been test in vivo. The lack of these data can cause one to wonder whether these approaches were tried and were ambiguous? At minimum, this issue should be discussed.

2) A bit surprising is the fact that the authors did not perform any myogenic marker studies. For example, are the PF3-low cells higher or lower expressing for early markers, such as MyoD1, or late markers, such as Myogenin or Mef2. It may not be PF3 per se, but these critical myogenic transcription and differentiation factors are true underpinnings. These studies need to be performed.

3) The authors spend a significant portion of their discussion looking towards findings from other neoplasms (Melanoma, Ewing Sarcoma) that might correlate with their finding. Given the perception that the manuscript overall reads as more descriptive, the discussion as currently configured aligns with the notion that perhaps the authors don't have clear translation to RMS, and are looking to other disease to help make biologic connections?

Minor Considerations:

1) The labeling in the plots shown in figure 1C is too small for viewing.

2) It is unclear in figure 2B whether the P-value of less than 0.5 applies to both studies or one

versus the other.

3) Since Figure 2A is an extension of the previous cell sorting the authors performed, this panel is not necessary.

Detailed point-by-point response to reviewers:

We appreciate the reviewer's thorough review of our manuscript and positive feedback. Responses are marked in blue. Major changes were underscored in the revised manuscript.

Reviewer # 1:

Summary: Rhabdomyosarcoma (RMS) is a soft tissue sarcoma in children and teens. There are two major subtypes. A Fusion negative subtype where *RAS* mutations are generally present in tumors with poor outcome and Fusion positive subtype is driven by a fusion oncogene containing either PAX3 or PAX7 fused to the FOXO1. While in FN-RMS tumors cell heterogeneity and the presence of cancer stem cells has been previously defined; however, in FP-RMS differences in tumorigenicity between sub-populations or tumors cell heterogeneity has not been clearly delineated and remains an open question with implications in how this tumor subtype propagates and responds to therapy.

Work by Keller and colleagues using a mouse model of FP-RMS in which the PAX3FOXO fusion oncogene is expressed along with YFP from the endogenous PAX3 promoter in Trp53^{-/-} animals. These studies have shown that there is heterogeneity in the expression of the fusion oncogene which is expressed at higher levels in the G2 phase of the cell cycle. This study by Regina et al., which uses low passage mouse FP-RMS cells generated from the murine model adds significantly to defining effects of differential levels of the PAX3FOXO oncogene on the ability to propagate tumor in vitro and in vivo. Both the in vitro clonal analyses and in the in vivo mouse xenograft experiments clearly show that the PAX3FOXO1 low cells have a significantly higher tumorigenic potential compared to PAX3FOXO Hi cells. Importantly, when high and low cells are sorted and plated they give rise to clonal growth and tumors with mixed populations similar to the unsorted parental cell line. The authors also confirm that there is heterogeneity in the fusion oncogene in human FP-RMS cell lines and low passage Patient derived xenografts. Additionally, the authors show that the proportion of cells expression PAX3FOXO1 fusion levels in the U23674 cell line can change in response to changes in the medium, the substratum, density and perturbation with vincristine or dactinomycin. Overall, this study is able to assess an important question in the field about the effect of heterogeneity in PAX3FOXO1 expression and effects on tumorigenicity. However, more broadly this study also highlights some of the challenges in treating tumors with heterogeneous tumor populations that can respond differently to the current standard of care.

We thank the reviewer for the careful review of our manuscript and positive feedback.

Comments:

1. While a single murine derived FP-RMS cell line U23674 used in all the analyses some of the major effects on heterogeneity in PAX3FOXO are not observed in the human cell lines; while this difference may be due to the human FP-RMS changing from being in culture for many years, it could also be that the U23674 cell line is an outlier. Can the authors perform similar in vitro assays in Figure 3 with the other mouse cell line U21459 i.e. growth in a) high glutamine b) laminin or matrigel c) higher densities of cells d) treatment with vincristine and dactinomycin.

We appreciate the reviewer's comment and performed analogous experiments using the mouse Myf6Cre+/-, Pax3:Foxo1+/+, p53-/- RMS cell line U21459 (please see figure S4).

Similar to what was observed in U23674 cells, higher glutamine levels, culture at higher cell densities and vincristine exposure led to an increase in the percentage of YFP^{high}/ P3F^{high} U21459 cells. Culture on a Matrigel matrix and Dactinomycin decreased the percentage of YFP^{high}/ P3F^{high} U21459 cells. Changes in glucose concentrations in the medium did not affect the percentage of YFP^{high}/ P3F^{high} U21459 cells.

2. There is a paradox associated with the data in Figure 3 and Figure 4. PAX3FOXO YFP high cells are more proliferative while PAX3FOXO low/negative cells are slow cycling but more clonogenic and tumorigenic. However, it appears that when the cells are stressed or allowed to overgrow which is usually associated with slower growth there is an increase in PAX3FOXO YFP high cells. Thus, this data can also suggest that when exposed to stress high PAX3FOXO protects cells from apoptosis by inducing a G2M block in cycle rather than the cells actually proliferating.

Can the authors perform Cell cycle along with apoptosis analyses comparing PAX3FOXO YFP high to low cells in U23674 cell grown in a) high glutamine b) higher densities of cells c) treatment with vincristine and dactinomycin. This will help determine if the high PAX3FOXO state is the one that is able to survive treatment while the low PAX3FOXO state then allows invasion migration and tumorigenic seeding.

We appreciate the reviewer's suggestions and evaluated cell cycle and apoptosis in U23674 cells using Hoechst staining (please see figure S5). Cells were grown in DMEM containing 25 mM glucose and 4 mM glutamine at increasing cell densities. Cells plated at intermediate densities were also exposed to vincristine, dactinomycin, carrier solution (NaCl 0,9%) and low-glutamine conditions (25 mM glucose, 0.05 mM glutamine).

Our analyses revealed reduced absolute numbers of YFP^{low} and YFP^{high} cells exposed to vincristine and low-glutamine conditions; dactinomycin exposure only resulted in a trend towards lower absolute numbers of YFP^{high} cells. The overall distribution of YFP^{high} and YFP^{low} cells across cell cycle phases remained the same for all conditions. Generally, YFP^{low}/ P3F^{low} cells included more cells in G0/ G1 stages compared to YFP^{high}/ P3F^{high} cells. Changes in absolute cell numbers correlated with higher percentages of apoptotic YFP^{low} and YFP^{high} cells exposed to vincristine and a lower percentage of G2/M YFP^{high} cells cultured in low-glutamine conditions. Taken together, our observations do not indicate that high P3F levels protect cells from stress-induced apoptosis by inducing a G2/M block.

Data were included in the revised manuscript.

3. The data provided in figure 4 A and B do not match the text. In the plots provided it appears that PAX3FOXO YFP hi cells have significantly lower Necrotic/late apoptotic cells compared to unsorted or PAX3FOXO YFP low cells. Can the authors comment or provide new plots.

The reviewer rightfully points out that figure 4A suggests that there are lower percentages of necrotic/ late apoptotic Annexin-V+/7AAD+ cells within the P3F^{high}/YFP^{high} subset of U23674 cells. We note that there was a certain level of variability among samples and among experiments. Differences in the percentages of

necrotic/ late apoptotic cells did not reach statistical significance ($p=0.09$), which was indicated in Figure 4B and the text.

The text of the manuscript was revised to account for the trend towards higher percentages of necrotic/ late apoptotic Annexin-V+/7AAD+ cells P3F^{high}/YFP^{high} cells.

4. In Figure 4E-G can the authors better explain the rationale for switching from experiments with sorted YFP^{hi} and YFP^{low} cells to experiments with knocking down PAX3FOXO with siRNA. For example, are these cells mostly in the G0/G1 phase of the cell cycle, and more sensitive to vincristine treatment and more tumorigenic/ clonogenic?

As YFP/P3F expression in individual U23674 cells was not stable and fluctuated over time, we chose to investigate the adhesion capacity of YFP^{low}/ P3F^{low} and YFP^{high}/ P3F^{high} cells using siRNA-mediated downregulation of P3F expression. This was stated in the revised manuscript.

We agree that it is clear that YFP^{low}/P3F^{low} cells contain a substantially higher proportion of G0/G1 cells than YFP^{high}/P3F^{high} cells (please see figure 3D and figure S5A-B). We also agree that this is a possible explanation for their higher tumorigenic/ clonogenic capacity. This was emphasized in the discussion of the revised manuscript.

Reviewer #2:

Life Science Alliance: Negative correlation of single-cell PAX3:FOXO1 expression with tumorigenicity in rhabdomyosarcoma (ID# LSA-2020-01002-T).

This manuscript by Regina et al describes differing phenotypes between PAX3-FOXO1 (PF3) RMS cells expressing high levels and low levels of PF3. PF3-RMS is a deadly disease and an important clinical problem. The authors describe that RMS cells with low levels of PF3 produce a cellular phenotype characterized by more efficient adhesion and migration, which correlate with higher tumor propagating frequencies. The authors then go on to show that exposure to agents that disrupt the cytoskeleton reverse enhanced migration and adhesion of RMS cells. The authors therein conclude that heterogeneous expression of PF3 at the single cell level may provide a critical advantage during tumor progression.

While this manuscript is well communicated and the authors have provided data consistent with their conclusions, the manuscript in its present form is unfortunately rather descriptive, and thus lacks true mechanistic insight and clinical translation to human RMS - and is not yet ready for publication in LSA. The reasons for this assessment from this reviewer include the following:

1) Unfortunately, one major weakness of this manuscript is the lack of in vivo data. Other than an early xenograft study, the authors do not provide any in vivo data to augment their findings in the latter two-thirds of the manuscript. For example, different matrices that were used in the in vitro studies (or equivalents) and/or the therapeutics used on cells could have been test in vivo. The lack of these data can cause one to wonder whether these approaches were tried and were ambiguous? At minimum, this issue should be discussed.

We agree with the reviewer that the lack of in vivo data on the effects of actin-depolymerizing agents on the tumorigenic potential of RMS cells is a weakness of this manuscript. This will be addressed in future studies.

We would like to communicate that the in vivo experiments were not performed yet.

2) A bit surprising is the fact that the authors did not perform any myogenic marker studies. For example, are the PF3-low cells higher or lower expressing for early markers, such as MyoD1, or late markers, such as Myogenin or Mef2. It may not be PF3 per se, but these critical myogenic transcription and differentiation factors are true underpinnings. These studies need to be performed.

We appreciate this important comment and explored differences in expression of myogenic regulatory factors between YFP^{high}/ P3F^{high} and YFP^{low}/ P3F^{low} cells. Significantly higher levels of myoblast determination protein 1 (MyoD1) and a trend towards higher expression of paired-box transcription factor 7 (PAX7) and myogenic factor 5 (Myf5) in YFP^{high}/ P3F^{high} cells were noted (please see figure S9). This was discussed in the revised manuscript.

3) The authors spend a significant portion of their discussion looking towards findings from other neoplasms (Melanoma, Ewing Sarcoma) that might correlate with their finding. Given the perception that the manuscript overall reads as more descriptive, the discussion as currently configured aligns with the notion that perhaps the authors don't have clear translation to RMS, and are looking to other disease to help make biologic connections?

The discussion of published observations in melanoma and Ewing sarcoma cells was shortened in the revised manuscript.

Minor Considerations:

1) The labeling in the plots shown in figure 1C is too small for viewing.

The font of the labels in figure 1C was increased.

2) It is unclear in figure 2B whether the P-value of less than 0.5 applies to both studies or one versus the other.

We believe that the reviewer's comment refers to figure 3b (not 2b), and we apologize for the lack of clarity. The panel was revised to demonstrate that the p-value < 0.5 referred to the differences in tumor-propagating capacity between P3F^{high}/YFP^{high} and P3F^{low}/ YFP^{low} cells.

Figure 2b also clearly indicates significance levels.

3) Since Figure 2A is an extension of the previous cell sorting the authors performed, this panel is not necessary.

Panel A was removed from figure 2 and added to figure S1.

June 11, 2021

RE: Life Science Alliance Manuscript #LSA-2020-01002-TR

Dr. Simone Hettmer
University Medical Center Freiburg
Pediatric Hematology and Oncology
Mathildenstrasse 1
Freiburg 79106
Germany

Dear Dr. Hettmer,

Thank you for submitting your revised manuscript entitled "Negative correlation of single-cell PAX3:FOXO1 expression with tumorigenicity in rhabdomyosarcoma.". We would be happy to publish your paper in Life Science Alliance pending final revisions necessary to meet our formatting guidelines.

- please make sure the author order in your manuscript and our system match
- please be sure to add all Authors in an Author Contributions section in your main manuscript text
- please consult our manuscript preparation guidelines <https://www.life-science-alliance.org/manuscript-prep> and make sure your manuscript sections are in the correct order. Please move your main, supplementary figure, and table legends to the main manuscript text after the references section
- we encourage you to revise the figure legends for figures S3, S6, S12 such that the figure panels are introduced in an alphabetical order
- remove the label of panel A from the actual figures S7 and S8, as they have only one panel
- there is a callout for Figure S13R although the actual figure has only A and B panels. Please correct
- please add a callout for Figures 6D; 7A-L, N, S, T; S5E; S10E and H; S14A-N, and P to your main manuscript text
- please revise the inset position in Figure S10G and J; S14R upper panel so that they match the zoomed-in parts

FIGURE CHECKS:

- missing scale bars for 6I; 7S, T, U; S10K; S14V, Y, X

To avoid unnecessary delays in the acceptance and publication of your paper, please read the

following information carefully.

A. FINAL FILES:

B. MANUSCRIPT ORGANIZATION AND FORMATTING:

Sincerely,

Reviewer #1 (Comments to the Authors (Required)):

The authors have addressed all the concerns raised in the initial review and I would recommend this manuscript for publication.

Reviewer #2 (Comments to the Authors (Required)):

The authors have responded to and satisfied all of my previous comments. I approve of the publication of this manuscript in LSA.

June 17, 2021

RE: Life Science Alliance Manuscript #LSA-2020-01002-TRR

Dr. Simone Hettmer
University Medical Center Freiburg, Pediatric Hematology and Oncology
Mathildenstrasse 1
Freiburg 79106
Germany

Dear Dr. Hettmer,

Thank you for submitting your Research Article entitled "Negative correlation of single-cell PAX3:FOXO1 expression with tumorigenicity in rhabdomyosarcoma.". It is a pleasure to let you know that your manuscript is now accepted for publication in Life Science Alliance. Congratulations on this interesting work.

DISTRIBUTION OF MATERIALS:

Again, congratulations on a very nice paper. I hope you found the review process to be constructive and are pleased with how the manuscript was handled editorially. We look forward to future exciting submissions from your lab.

Sincerely,
